# Paneth cell α-defensin misfolding correlates with dysbiosis and ileitis in Crohn's disease model mice

Yu Shimizu[1,2] , Kiminori Nakamura[1,2], Aki Yoshii[1], Yuki Yokoi[1,2] , Mani Kikuchi[2], Ryuga Shinozaki[1], Shunta Nakamura[1], Shuya Ohira[1] , Rina Sugimoto[1], Tokiyoshi Ayabe[1,2]

**Crohn's disease (CD) is an intractable inflammatory bowel disease, and dysbiosis, disruption of the intestinal microbiota, is associated with CD pathophysiology. ER stress, disruption of ER homeostasis in Paneth cells of the small intestine, and α-defensin misfolding have been reported in CD patients. Because α-defensins regulate the composition of the intestinal microbiota, their misfolding may cause dysbiosis. However, whether ER stress, α-defensin misfolding, and dysbiosis contribute to the pathophysiology of CD remains unknown. Here, we show that abnormal Paneth cells with markers of ER stress appear in SAMP1/YitFc, a mouse model of CD, along with disease progression. Those mice secrete reduced-form α-defensins that lack disulfide bonds into the intestinal lumen, a condition not found in normal mice, and reduced-form α-defensins correlate with dysbiosis during disease progression. Moreover, administration of reduced-form α-defensins to wild-type mice induces the dysbiosis. These data provide novel insights into CD pathogenesis induced by dysbiosis resulting from Paneth cell α-defensin misfolding and they suggest further that Paneth cells may be potential therapeutic targets.**

## Introduction

The intestinal tract harbors an immense number of bacteria, the intestinal microbiota, which are involved in many aspects of host physiology, that includes energy metabolism (1), immune system regulation (2), and nervous system development (3). Imbalance of the intestinal microbiota, termed dysbiosis, is associated with many diseases, including chronic lifestyle diseases such as obesity and diabetes, immunological disorders, and nervous system diseases (4). α-Defensins, a major family of mammalian antimicrobial peptides, are known regulators of the intestinal microbiota. These ~4-kD basic peptides are characterized by evolutionarily conserved Cys residue positions that are invariantly spaced to form disulfide

bonds between $Cys^I$-$Cys^{VI}$, $Cys^{II}$-$Cys^{IV}$, and $Cys^{III}$-$Cys^V$ (5). In the intestinal epithelium, α-defensins occur only in intracellular dense-core secretory granules of Paneth cells, one of the major terminally differentiated lineages of the small intestine. Paneth cells, which reside at the base of the crypts of Lieberkühn, release secretory granules that are rich in α-defensins, termed cryptdins (Crps) in mice and HD5 and HD6 in human, in response to bacteria and other stimuli at effective concentrations, thereby contributing to enteric innate immunity (6, 7, 8, 9, 10, 11). Also, Paneth cell α-defensins contribute to regulating the composition of the intestinal microbiota in an activity-dependent manner in vivo and affecting development of host-adaptive immunity (12). Furthermore, oral administration of Crp4 prevents severe dysbiosis in mouse graft-versus-host disease (13, 14), indicating that Paneth cell α-defensins secreted into the intestinal lumen contribute not only to innate immunity but also to maintenance of intestinal homeostasis by regulating the intestinal microbiota (15, 16).

Recently, a relationship has been revealed between the intestinal microbiota and the pathophysiology of Crohn's disease (CD) (17). CD is a chronic inflammatory bowel disease (IBD) that may affect the entire gastrointestinal tract, especially the terminal ileum, with chronic inflammation and ulceration (18). The number of patients with CD has been increasing continuously worldwide, including Europe, the Americas, and Asia (18, 19, 20). Although a complete picture of CD pathogenesis is lacking, there is consensus that dysbiosis and dysregulated immune responses to the intestinal microbiota play important roles (18). Moreover, both genetic factors consisting of more than 160 susceptibility loci (21), as well as environmental factors such as overuse of antibiotics (22) and adoption of "Westernized diets" (23) have been reported as CD risk factors, and these factors are suggested to induce pathophysiology of CD via dysbiosis (24).

Evidence shows that certain Paneth cell defects are involved in CD onset and pathophysiology. Paneth cells continuously synthesize high levels of secretory proteins in the ER and are susceptible to ER stress and failure to maintain ER homeostasis because of accumulation of misfolded proteins (25). Several genes involved in resolution of ER stress affect CD susceptibility and

[1]Innate Immunity Laboratory, Graduate School of Life Science, Hokkaido University, Hokkaido, Japan   [2]Department of Cell Biological Science, Faculty of Advanced Life Science, Hokkaido University, Hokkaido, Japan

Correspondence: ayabe@sci.hokudai.ac.jp

deletions or mutations of such gene. For example, unfolded protein response (UPR)–related genes *XBP1* (26) and *AGR2* (27), autophagy-related genes *ATG16L1* (28), *IRGM1* (29), and *LRRK2* (30) cause Paneth cell abnormalities in granule morphology and cellular localization in mouse models. In CD patients with mutations in *XBP1* and *ATG16L1*, Paneth cell abnormalities occur, which are similar to those observed in genetically deficient mice (25). Moreover, several studies have identified relationships between Paneth cell ER stress and disruptions of the intestinal microbiota in CD. The appearance rate of Paneth cells with abnormal granule morphology in CD patients is associated with dysbiosis that is characterized by reduction of diversity and decrease of anti-inflammatory bacteria such as *Faecalibacterium* (31). CD patients positive for the ER stress marker GRP78 in Paneth cells harbor greater numbers of CD-associated enteroinvasive *Escherichia coli* (*E. coli*) compared with patients that lack GPR78-positive Paneth cells (32). Although these findings suggest the involvement of ER stress, Paneth cell dysfunction, and dysbiosis in CD pathophysiology, causal relationships between these factors remain to be demonstrated.

Abnormal posttranslational modifications occur in proteins synthesized in ER-stressed cells, including the misfolding of disulfide bonds (33, 34). For example, oxidized-form Crps (oxCrps) containing three intramolecular disulfide bonds elicit strong bactericidal activities against pathogenic bacteria and minimal or no bactericidal activities against commensal bacteria in vitro. In contrast, disulfide-null reduced-form Crps (rCrps) kill both pathogens and commensals in vitro (35). Furthermore, CD patients have reduced-form HD5 in their ileal tissue, a condition not detected in healthy subjects (36). Taken together, we hypothesized that Paneth cell ER stress would promote biosynthesis of reduced-form α-defensins because of disulfide bond misfolding and that secretion of reduced-form α-defensins into the intestinal lumen would disrupt the spectrum of bactericidal activities against commensals and induce dysbiosis, resulting in the onset of CD pathophysiology.

SAMP1/YitFc is a mouse model of CD that develops a spontaneous ileitis which closely resembles in CD patients (37, 38, 39). Here, we show that increased numbers of abnormal Paneth cells with disrupted cellular localization appear with progression of ileitis in mice. We demonstrate that the abnormal Paneth cells undergo ER stress and that they secrete rCrps into the intestinal lumen. Furthermore, SAMP1/YitFc mice with developed ileitis exhibit dysbiosis characterized by loss of diversity, decreased occupancy of Lachnospiraceae and Ruminococcaceae and increased occupancy of Bacteroidaceae and Rikenellaceae. Furthermore, the quantity of rCrps secreted into the lumen has a strong positive correlation with dysbiosis and disease progression. Results from this model system suggest a novel mechanism of CD pathogenesis based on α-defensin misfolding in Paneth cells.

# Results

### Increased numbers of abnormal Paneth cells during disease progression in SAMP1/YitFc mice

First, histological analysis with hematoxylin and eosin (HE) staining was conducted to assess the relationship between disease progression and

Paneth cells in SAMP1/YitFc mice, which develop spontaneous ileitis by 20 wk of age which resembles that of CD patients (37). In SAMP1/YitFc mice, progression of ileitis was observed from 4 to 20 wk characterized by inflammatory cell infiltration, crypt elongation, thickening of muscle layer, and crypt abscess (Figs 1A and S1). Inflammatory scores were significantly elevated from 10 to 20 wk (Table S1 and Fig 1B; 4 wk: 0.19 ± 0.08, 10 wk: 1.00 ± 0.22, and 20 wk: 2.74 ± 0.49). At 10 and 20 wk, abnormal Paneth cells were observed not only at the base of crypts but also in the upper part of crypts and on villi, where Paneth cells are absent in controls (Fig 1A). Also, the number of abnormal Paneth cells increased significantly from 4 to 10 wk (Fig 1C; 4 wk: 3.54 ± 0.16, 10 wk: 6.34 ± 0.63, and 20 wk: 8.30 ± 0.90 cells/crypt-villus axis). At 10 and 20 wk, eosinophilic granule–positive cells also stained positive for Alcian blue which detects mucus normally produced by goblet cells (Fig 1D). These findings indicate that the abnormal Paneth cells may be intermediate cells which have properties common to both Paneth and goblet cells (40). Moreover, co-expression of Crps and Muc2, a major constituent of mucus produced by goblet cells, was observed throughout the crypt-villus axis of SAMP1/YitFc mice at 20 wk, but Crps/Muc2 double-positive cells were not detected in controls (Fig 1E). Furthermore, a strong positive correlation was shown between the number of eosinophilic granule–positive abnormal Paneth cells and the inflammatory scores of individual SAMP1/YitFc mice (Fig 1F). Thus, the increase in abnormal Paneth cells was associated with disease progression in SAMP1/YitFc mice. To address possible mechanisms of abnormal Paneth cell accumulation in the upper crypts and villi, EphB2 expression was analyzed by immunofluorescent staining. Along with disease progression, the expression of EphB2, which is known to define the position of Paneth cells at the base of the crypt (41) was decreased in crypts of abnormal Paneth cells. No EphB2 expression was observed in abnormal Paneth cells on villi or in the crypts at 20 wk (Fig S2), suggesting that the decreased expression of EphB2 accompanying disease progression allows for the transfer of abnormal Paneth cells onto the villi.

### Abnormal Paneth cells exhibit ER stress

To observe abnormal SAMP1/YitFc mouse Paneth cells at higher resolution, transmission electron microscopy (TEM) was performed. Paneth cells in control mice had electron dense granules of uniform size in the apical region (Fig 2A). In contrast, abnormal Paneth cells of SAMP1/YitFc mice contained granules of non-uniform size with electron-lucent halos in the granule periphery (Fig 2B). These abnormalities in Paneth cell granule morphology were more evident and prominent in 20-wk SAMP1/YitFc mice than in 4-wk animals. In addition, the ER structure of control mouse Paneth cells displayed orderly, stacked ER cisternae around nuclei, but SAMP1/YitFc mouse Paneth cells showed swollen, disordered ER structures, which are characteristic of ER stress, from 4 to 20 wk. These ER abnormalities were observed in all Paneth cells randomly selected from three 20-wk SAMP1/YitFc mice (Fig S3). Paneth cell abnormalities were quantified using TEM images (Fig S4). In SAMP1/YitFc mice, the number of granules increased significantly at 20 wk compared with 4 wk (Fig 2C; 4 wk: 19.0 ± 2.2 and 20 wk: 28.1 ± 2.0 granules/Paneth cell). In addition, ER luminal diameter increased significantly at 20 wk compared with 4 wk (Fig 2D; 4 wk: 0.054 ± 0.005 and 20 wk: 0.097 ± 0.002 µm). These results prompted us to analyze the expression of UPR-related molecules, which reflect ER stress

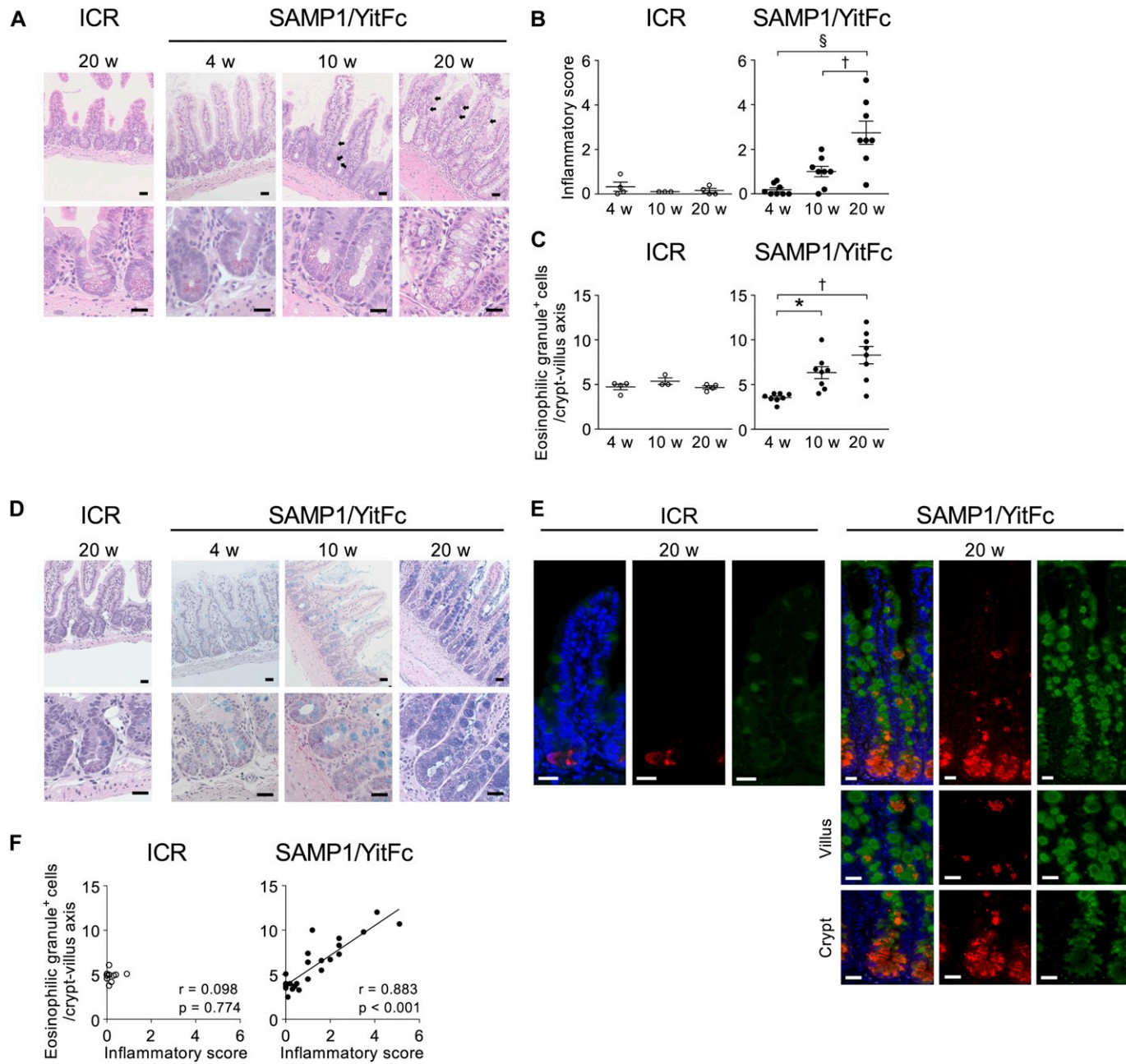

**Figure 1.  Abnormal Paneth cells increase during disease progression in SAMP1/YitFc mice.**
**(A)** Representative HE staining images of ileal tissues. Abnormal Paneth cells in upper crypts and on villi are indicated by black arrows. **(B)** Inflammatory scores (n = 3–4/each week in ICR mice, n = 8/each week in SAMP1/YitFc mice). **(C)** The number of eosinophilic granule–positive cells. **(D)** Representative HE–Alcian blue staining images of ileal tissues. **(E)** Representative immunofluorescent staining images for Muc2 (green) in ileal tissues. Crps (red) and DAPI (blue). **(F)** Correlation analysis between the number of eosinophilic granule–positive cells and inflammatory scores in each mouse at 4, 10, and 20 wk. In (A, D, E), scale bars indicate 20 μm. Error bars represent mean ± SEM. **(B, C, F)** Statistical significance was evaluated by one-way ANOVA followed by Tukey's post hoc test in (B, C), and Pearson's correlation coefficients test in (F). *P < 0.05, †P < 0.01, §P < 0.001.

intensity, in isolated ileal crypts (Fig 2E). Results showed that, although pIRE1α expression was not statistically different between 4 and 20 wk (20-wk ICR mice versus 4-wk SAMP1/YitFc mice versus 20-wk SAMP1/YitFc mice: 0.45 ± 0.07 versus 0.65 ± 0.11 versus 0.82 ± 0.13) and the expression of ER stress markers, ATF4 (0.69 ± 0.07 versus 0.48 ± 0.03 versus 1.20 ± 0.24), cleaved-ATF6 (0.69 ± 0.07 versus 0.48 ±

0.03 versus 1.20 ± 0.24), and GRP78 (0.44 ± 0.07 versus 0.18 ± 0.06 versus 0.85 ± 0.29), in ileal crypts was significantly increased in SAMP1/YitFc mice at 20 wk compared with those at 4 wk along with disease progression (Fig 2F). To clarify whether ER stress in the crypts determined by Western blot occurs in Paneth cells, immunofluorescent staining of the expression of ER stress markers,

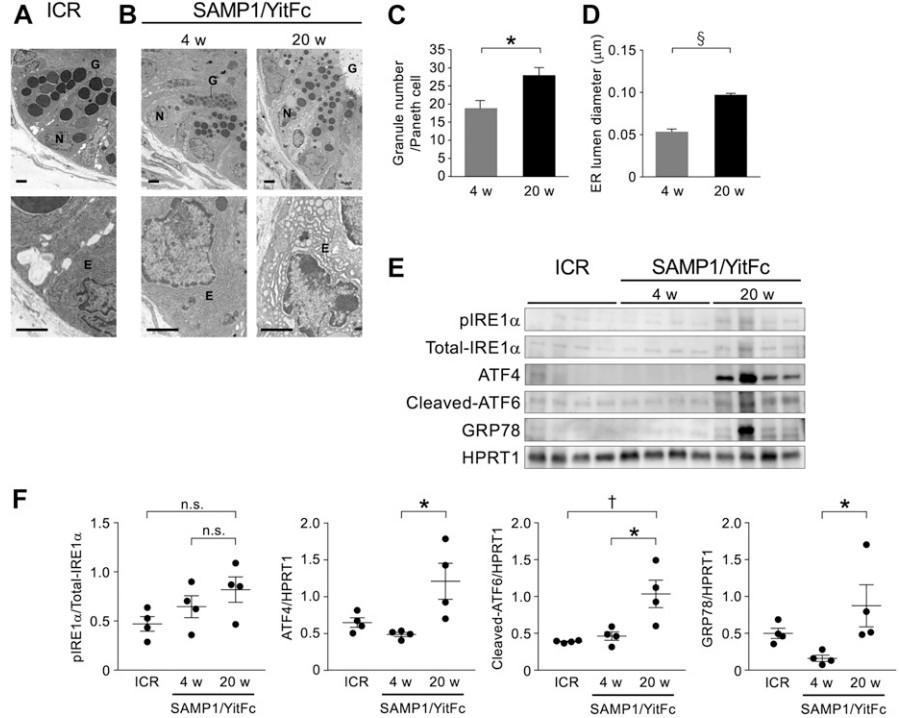

**Figure 2. Abnormal Paneth cells show ER stress.**
**(A, B)** Representative transmission electron microscopy images of Paneth cells at the base of ileal crypts in (A) ICR and (B) SAMP1/YitFc mice. Scale bars indicate 2 μm. **(C, D)** Quantitative analysis of (C) granule number and (D) ER lumen diameter in Paneth cells (n = 3/each week for SAMP1/YitFc mice). For the measurements, three Paneth cells were randomly selected from each mouse. **(E)** SDS–PAGE Western blot analysis of ER stress markers, pIRE1α, ATF4, cleaved-ATF6, and GRP78 in ileal crypts (n = 4/each group). Total-IRE1α and HPRT1 was used as loading control. **(F)** Relative expression level of ER stress markers calculated from the band intensity. Error bars represent mean ± SEM. **(C, D, F)** Statistical significance was evaluated by t test in (C, D), and one-way ANOVA followed by Tukey's post hoc test in (F). *P* < 0.05 was considered statistically significant. *$P$ < 0.05, †$P$ < 0.01, §$P$ < 0.001. E, ER; G, granules; N, nucleus; n.s., not significant.

GRP78 and calreticulin, downstream of ATF6, which is known as representative of ER stress sensors, in ileal tissues of both ICR and SAMP1/YitFc mice were performed. In SAMP1/YitFc mice at 20 wk, the expression of both GRP78 and calreticulin was remarkably increased in abnormal Paneth cells, in contrast to low expression in ICR mice (Fig S5A and B). Furthermore, we analyzed expression of transcription factor, MIST1, which is known to suppress ER stress and is specifically expressed in Paneth cells of the intestinal epithelium (42). By immunofluorescent staining, MIST1 was strongly expressed in the nucleus of Paneth cells in 20-wk ICR mice. In sharp contrast, MIST1 expression was dramatically diminished at 20 wk compared with 4 wk in SAMP1/YitFc mouse Paneth cells (Fig S5C). These results indicated that ER stress in crypts of SAMP1/YitFc mice occurs in Paneth cells during disease progression.

### Abnormal Paneth cells produce reduced-form cryptdins

Because abnormal Paneth cells exhibited ER stress, we next examined whether rCrps are produced in these cells because of misfolding of disulfide bonds. Crps consist of multiple isoforms with extensive similarities in amino acid sequences (43). We purified Crps from small intestinal tissues and analyzed their tertiary structures by acid urea (AU)-PAGE Western blot using a pan-Crp antibody, which detects both ox and rCrp isoforms (Fig S6A). In AU-PAGE, oxCrp1 showed the lowest mobility among oxCrps and oxCrp1 showed higher mobility than all rCrps (Fig S6B). Thus, using oxidized and reduced forms of chemically synthesized Crp1 (44) as markers for evaluating tertiary structure, Crps in small intestinal tissue detected with lower mobilities than oxCrp1 were judged to be rCrps, whereas peptides with higher mobilities were judged to be oxCrps. rCrps were detected in 20-wk SAMP1/YitFc mice

but were not found in age-matched controls or in 4 wk SAMP1/YitFc mice (fractions 3–10 in Fig 3). Multiple bands of rCrps with different mobilities were detected in these fractions, suggesting that numerous isoforms were misfolded. In addition, levels of oxCrps increased in 20 wk (fraction 1 and 2 in Fig 3) compared with 4-wk SAMP1/YitFc mice and controls. These results indicate that abnormal Paneth cells produce rCrps, which are not detected in the normal state, coincident with progression of ER stress.

### Reduced-form cryptdins in abnormal Paneth cells are secreted into the intestinal lumen and relate to disease progression

To clarify whether the rCrps that accumulate in abnormal Paneth cells are secreted into the intestinal lumen, we quantified rCrps in mouse feces. oxCrps are known to resist degradation by digestive enzymes in vitro, in contrast to Crp mutants that lack one or more disulfide bonds and are easily degraded (45). Using this property of Crps, mouse fecal extracts were treated with trypsin, then Crps in feces were separated and quantified by Western blots after denatured tricine SDS–PAGE separations. We then quantified ox and rCrps by comparing the quantity of Crps in trypsin-treated and untreated samples (Fig 4). Distinguishing ox and rCrps directly in fecal extracts by AU-PAGE Western blots was not feasible because electrophoresis of the extracts was inhibited, perhaps by fecal contaminants. Trypsin treatment of a synthetic, rCrp1 peptide standard caused gel bands to disappear completely, in contrast to oxCrp1 bands, which were unaffected by trypsin exposure. Therefore, we implemented this method for measuring ox and rCrps based on the sensitivity to trypsin. In control and 4 wk of SAMP1/YitFc mice, band intensities of Crps detected in fecal extracts showed no significant changes resulting from trypsin

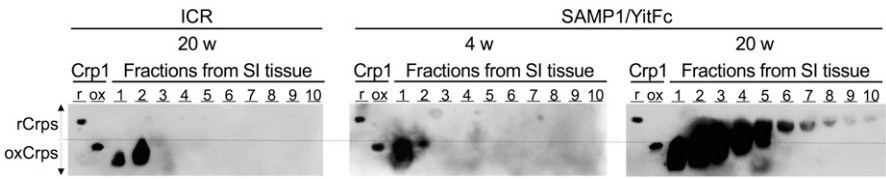

**Figure 3.  Reduced-form α-defensins accumulate in abnormal Paneth cells.**
Protein was extracted from full length of small intestinal (SI) tissue obtained from 10 ICR and 10 SAMP1/YitFc mice and fractionated by using preparative acid native-PAGE system. AU–PAGE Western blot analysis of Crps in each fraction was conducted. Fraction numbers denote the order of elution during fractionation. 200 ng of chemically synthesized ox and rCrp1 were used as markers for identifying the conformation of Crps in each fraction. Based on band motility, Crps detected in the mobility zone below oxCrp1 (below the dotted line) was judged as oxCrps, and Crps detected in the lower motility zone than that of oxCrp1 (above the dotted line) was judged as rCrps.

digestion. However, Crps levels in fecal extracts from 20 wk of SAMP1/YitFc mice were significantly decreased by trypsin digestion (Fig 4A and B), showing that rCrps produced by abnormal Paneth cells were secreted into the intestinal lumen and sensitive to proteolysis. Accordingly, we quantified ox and rCrps in feces on the basis of their sensitivity to trypsin. For example, Crps detected in fecal samples not subjected to tryptic digestion was considered to be total Crps, that is, the sum of ox and rCrps, and Crps detected in fecal samples treated with trypsin was identified as oxCrps. Therefore, values obtained by subtracting the quantity of Crps after trypsin treatment from that of samples not treated with trypsin were taken to be the amount of rCrps (Fig 4C). Levels of oxCrps in feces of SAMP1/YitFc mice approximately doubled at 20 wk compared with 4 wk (4 wk: 6.50 ± 1.46 and 20 wk: 11.76 ± 1.12 ng/500 μg feces). In sharp contrast, the quantity of rCrps in feces of 20-wk SAMP1/YitFc mice increased ~40-fold compared with 4-wk animals (4 wk: 0.23 ± 0.49 and 20 wk: 9.82 ± 3.12 ng/500 μg feces). Therefore, rCrps, which were almost undetectable before disease onset, are secreted into the intestinal lumen at high levels during disease progression. In control feces, the amount of oxCrps secreted

was unchanged during the first 20 wk, and rCrps was not detected. To test whether synthesis of rCrps is involved in IBD progression, correlation analyses between the quantity of ox and rCrps levels in feces and the inflammatory scores of individual SAMP1/YitFc mice were conducted (Fig 4D). A strong positive correlation existed between the levels of rCrps and inflammatory scores, but no significant correlation was observed between quantities of oxCrps and inflammatory scores. To clarify whether rCrps detected in feces of SAMP1/YitFc mice were secreted from abnormal Paneth cells, Paneth cell granule secretion was visualized and quantified using enteroids, three-dimensional cultures of small intestinal epithelial cells, including Paneth cells. Using carbachol (CCh) to induce granule secretion in enteroid Paneth cells, granule secretion ratio of the SAMP1/YitFc mouse–derived Paneth cells was equivalent to that of ICR mouse Paneth cells (Fig S7A; 56.55% ± 2.85% versus 46.97% ± 4.44%). In addition, we tested whether abnormal Paneth cells survived during and after secretion, that is, did not undergo cell death. After treatment of SAMP1/YitFc mouse enteroids with a fluorescent probe specific to active caspase-3/7 as reported previously (16), granule secretion was induced by CCh. No

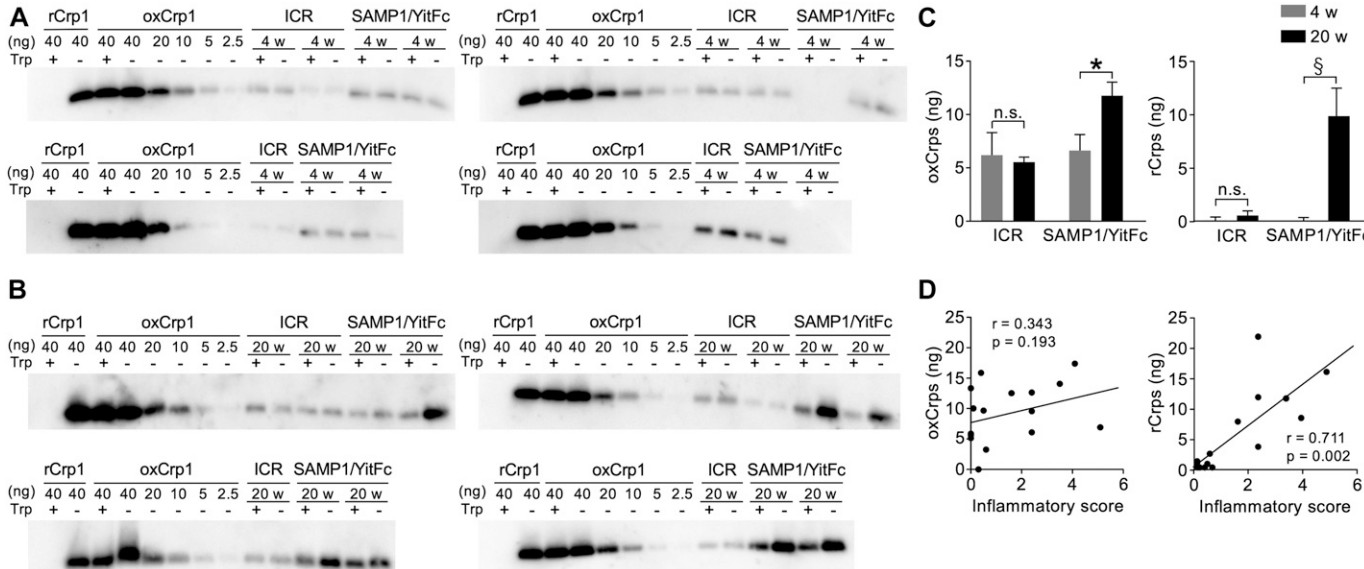

**Figure 4.  Reduced-form α-defensins are secreted into the intestinal lumen and associated with the disease progression.**
**(A, B)** Tricine SDS–PAGE Western blot analysis of Crps in fecal extracts of (A) 4 wk and (B) 20 wk (ICR mice; n = 6, SAMP1/YitFc mice; n = 8/week). Chemically synthesized ox and rCrp1 were used as positive controls. Each sample was treated with trypsin (Trp+) or PBS (Trp–). Band intensity of oxCrp1 Trp+ was approximately equivalent to that of oxCrp1 Trp–, whereas bands of rCrp1 Trp+ were hardly visible. Thus, Crps detected in Trp– were considered to be the sum of ox and rCrps, and Crps detected in Trp+ were considered as rCrps that escaped tryptic digestion. **(C)** The amount of ox and rCrps in fecal extracts. The amount of oxCrps in each sample was calculated from band intensity of Trp+, and the amount of rCrps was calculated from the difference between band intensity of Trp– and Trp+. **(D)** Correlation analysis between the quantity of fecal Crps and inflammatory scores of each SAMP1/YitFc mouse. Error bars represent mean ± SEM. **(C, D)** Statistical significance was evaluated by Mann–Whitney's U test in (C), and by Pearson's correlation coefficients test in (D). *P* < 0.05 was considered statistically significant. *P < 0.05, §P < 0.001.

cleaved caspase was detected in Paneth cells during or after secretion (Fig S7B and Video 1). These results indicated that secretion of rCrps by abnormal Paneth cells is further associated with the disease progression in SAMP1/YitFc mice.

### Dysbiosis occurs with disease progression in SAMP1/YitFc mice

Because rCrps has been known to elicit an abnormal bactericidal spectrum in vitro against commensal bacteria compared with oxCrps (35), we tested whether rCrps secreted into the intestinal lumen induce dysbiosis in SAMP1/YitFc mice. First, we determined whether dysbiosis occurs in SAMP1/YitFc mice using 16S ribosomal DNA (rDNA) metagenomic sequencing of the intestinal microbiota in fecal samples. Both phylogenic diversity (PD) whole tree (4 versus 20 wk: 15.61 ± 0.56 versus 13.64 ± 0.44) and operational taxonomic units (OTUs) (194.88 ± 7.40 versus 158.68 ± 7.01) $\alpha$-diversity indexes

were decreased significantly during disease progression (Fig 5A and B). In addition, significant decreases of Lachnospiraceae (34.37% ± 1.89% versus 24.86% ± 1.65%) and Ruminococcaceae (9.70% ± 0.83% versus 6.83% ± 1.00%) and increases of Bacteroidaceae (29.82% ± 1.75% versus 35.67% ± 1.71%) and Rikenellaceae (5.64% ± 0.96% versus 8.69% ± 1.05%) were observed at the family level in 20 wk compared with 4-wk mice (Fig 5C). At the genus level, a significant decrease of *Lachnospiraceae;Other* (6.23% ± 1.08% versus 3.36% ± 0.66%) and *Anaerotruncus* (0.74% ± 0.17% versus 0.38% ± 0.07%) and an increase of *Bacteroides* (29.82% ± 1.75% versus 35.67% ± 1.71%) were shown in 20 wk (Fig 5D). To clarify further whether the dysbiosis that occurred in 20-wk SAMP1/YitFc mice relates to disease progression, correlation analyses between the microbiota composition and individual inflammatory scores were performed. Negative correlations were observed between each $\alpha$-diversity index and inflammatory scores (Fig 5E and F). Moreover, at the

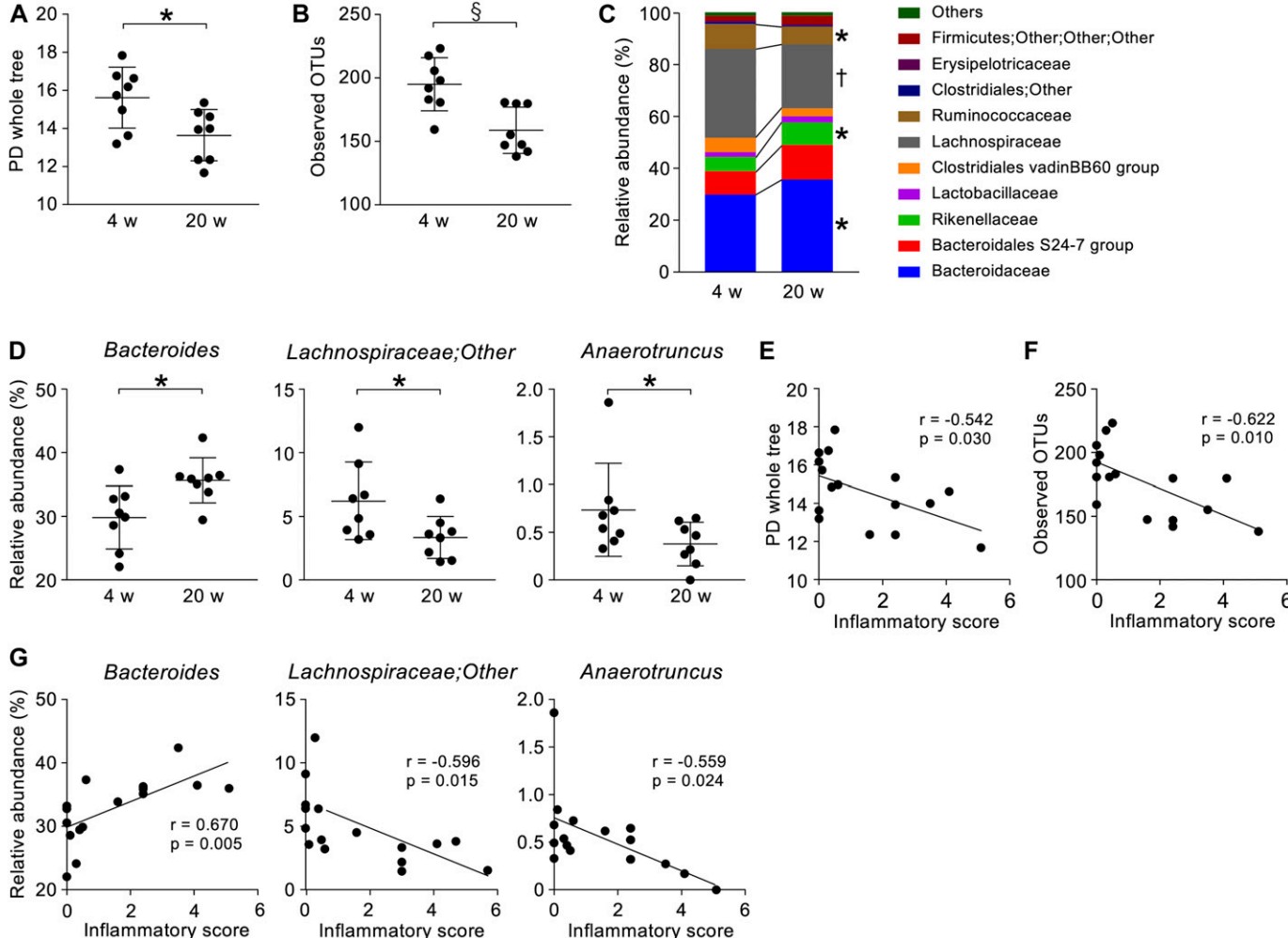

**Figure 5. Dysbiosis occurs along with pathological progression.**
**(A, B)** Two $\alpha$-diversity indexes, (A) PD whole tree and (B) observed OTUs in SAMP1/YitFc mice. **(C)** Stacked bar chart of relative abundance of each taxon in SAMP1/YitFc mice at the family level. **(D)** Dot plots of significantly changed taxa in SAMP1/YitFc mice at genus levels. **(E, F)** Correlation analysis between inflammatory scores and $\alpha$-diversify indexes of each SAMP1/YitFc mouse. **(G)** Correlation analysis between inflammatory scores and significantly changed taxa in SAMP1/YitFc mice at the genus level. Error bars represent mean ± SEM. **(A, B, C, D, E, F, G)** Statistical significance was evaluated by Mann–Whitney's U test in (A, B, C, D), and Pearson's correlation coefficients test in (E, F, G). $P < 0.05$ was considered statistically significant. *$P < 0.05$, †$P < 0.01$, §$P < 0.001$.

genus level, relative abundance of *Lachnospiraceae;Other* and *Anaerotruncus* correlated negatively and *Bacteroides* correlated positively with the inflammatory score (Fig 5G). There was no significant difference in β-diversity between ICR and SAMP1/YitFc mice at 4 wk, suggesting that ICR is a suitable control strain for the intestinal microbiota analysis. At 20 wk when disease has progressed in SAMP1/YitFc mice, the intestinal microbiota in ICR and SAMP1/YitFc mice showed significantly different composition (Fig S8). We next compared α-diversity and relative abundance of the intestinal microbiota in control mice between 4 and 20 wk. No significant change in α-diversity or in the relative abundance was detected in the genera that significantly shifted between 4 and 20 wk in SAMP1/YitFc mice (Fig S9). Thus, dysbiosis with decreased diversity and compositional changes in certain taxa accompanies disease progression in SAMP1/YitFc mice.

## Secretion of reduced-form Crps into the intestinal lumen correlates with dysbiosis

To clarify the relationship between secretion of rCrps into the intestinal lumen and the progression of dysbiosis in SAMP1/YitFc mice, correlation analysis between the quantity of rCrps in feces and α-diversity indexes of individual animals was conducted. A strong negative correlation was observed between the two α-diversity indexes and the amount of rCrps (Fig 6A and B). Furthermore, strong negative correlations were found between levels of rCrps and the occupancy of *Lachnospiraceae;Other* and *Anaerotruncus*, but there was a strong positive correlation with *Bacteroides* (Fig 6C). In contrast, no correlation was observed between the amount of oxCrps in feces, the diversity indexes, or bacterial composition of the fecal microbiota (Fig S10). In vitro, rCrp1 had significantly stronger bactericidal activity than oxCrp1 against

*Anaerotruncus colihominis* (*A. colihominis*), a commensal bacterium of the genus *Anaerotruncus* that induces regulatory T-cell differentiation in the intestine (Fig S11) (2). Furthermore, to exclude that dysbiosis could lead to Paneth cell defects, we analyzed the morphology of Paneth cells and secretion of rCrps after administering antibiotics to SAMP1/YitFc mice. The PCR of fecal 16S rDNA confirmed that intestinal bacteria could be eliminated completely by oral administration of antibiotics (Abx) for 6 wk (Fig S12B). The number of eosinophilic granule–positive cells, that is, abnormal Paneth cells, per villus–crypt axis in Abx-treated group was similar to water-treated mouse at 10 wk in SAMP1/YitFc mice (Fig S12C; water versus Abx-treated: 5.20 versus 4.25 ± 0.70 cells/crypt-villus axis). In addition, TEM analyses showed ER swelling in Paneth cells of Abx group, as in water-treated mice (Fig S12D; water versus Abx-treated: 0.078 versus 0.070 ± 0.001 μm). Furthermore, it was confirmed that rCrps were present in feces of both the Abx group and water-treated mice (Fig S12E; water versus Abx-treated: 8.06 versus 6.35 ± 1.33 ng). Taken together, data showed that Paneth cells are abnormal even in the absence of the intestinal bacteria during pathogenesis in SAMP1/YitFc mice. Finally, we further tested whether reduced-form Crps could change the intestinal microbiota using ICR mice. Because rCrp4 have been reported to be degraded by proteases in vitro (45) and considering that oral administration of rCrps may result in degradation in the stomach and loss of activity in the intestinal lumen, we administered rCrp1 rectally. The observed OTUs indicating α-diversity in the rCrp1 group were significantly decreased compared with the control group at day 4 (Fig S13B; control versus rCrp1-treated: 274.00 ± 4.16 versus 277.25 ± 8.64 at day 0; 292.67 ± 7.86 versus 249.25 ± 22.48 at day 2; 305.33 ± 3.18 versus 241.00 ± 20.06 at day 4). Furthermore, Lachnospiraceae (14.59% ± 2.83% versus 13.96% ± 0.48% at day 0; 18.97% ± 4.14% versus 10.61% ± 0.46% at day 2; 24.90% ± 6.48% versus 13.13% ± 1.21% at day 4) and Ruminococcaceae (6.28% ± 0.24% versus 6.34% ± 0.70% at day

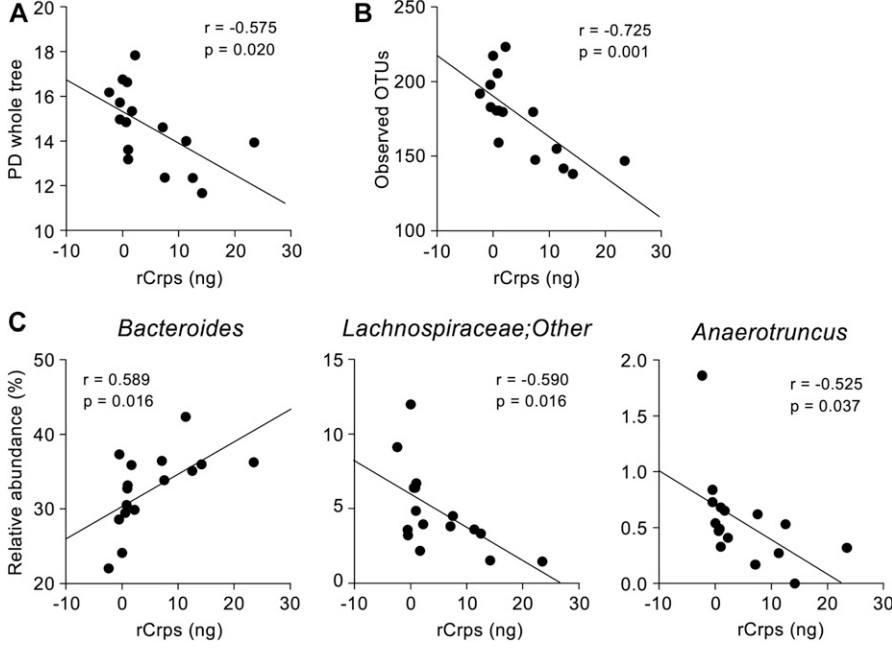

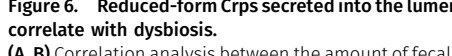

**Figure 6. Reduced-form Crps secreted into the lumen correlate with dysbiosis.**
**(A, B)** Correlation analysis between the amount of fecal Crps and α-diversity indexes of each SAMP1/YitFc mouse. **(C)** Correlation analysis between the quantity of fecal Crps and relative abundance of significantly changed taxa in SAMP1/YitFc mice. Statistical significance was evaluated by Pearson's correlation coefficients test. *P* < 0.05 was considered statistically significant.

0; 7.22% ± 0.85% versus 3.05% ± 0.62% at day 2; 8.51% versus 3.76% ± 0.89% at day 4), both which were decreased along with disease progression of SAMP1/YitFc mice (Fig 5C), were significantly decreased in the rCrp1 group, at day 4 and at day 2 and day 4, respectively (Fig S13C). In addition, there was a significant negative correlation between abundances of Ruminococcaceae and both the inflammatory score and the quantity of rCrps secreted (Fig S14). Although rectal administration of rCrp to ICR mice did not replicate the SAMP1/YitFc mouse enteropathy, we found that rCrp could modulate the fecal microbiota. Taken together, our results showed that rCrps secreted into the intestinal lumen could be involved in dysbiosis associated with disease progression in the SAMP1/YitFc mouse model of IBD.

# Discussion

Dysbiosis with reduced diversity observed in the SAMP1/YitFc mice is consistent with previous studies of the intestinal microbiota of CD patients from the Americas, Europe, and Japan (31, 46, 47, 48). Furthermore, decreases of both Lachnospiraceae and Ruminococcaceae along with disease progression shown here also have been reported in CD patients in the Americas and Japan (31, 48, 49). Because these bacteria produce butyrate (50), an inducer of regulatory T-cell differentiation (2), a decrease in Lachnospiraceae and Ruminococcaceae may lead to excessive adaptive immune responses in the intestine. Moreover, increased Bacteroidaceae, which increased in SAMP1/YitFc mice along with disease progression, have been reported in CD patients in the United Kingdom and Canada (46, 51). The Bacteroidaceae include some opportunistic pathogens (52), and their overgrowth may induce enteric mucosal inflammation. Certain Rikenellaceae, which also increased in SAMP1/YitFc mice, produce capnine, a sulfolipid inhibitor of the vitamin D receptor (53, 54). Because activation of the vitamin D receptor up-regulates transcription factors for antimicrobial peptides, including LL-37 (55) and α-defensins (56), and genetic deletion of the vitamin D receptor in intestinal epithelial cells leads to reduction of autophagy (57), increased Rikenellaceae may disrupt innate enteric immunity and result in epithelial cell dysfunction. The shifts of the intestinal microbiota in SAMP1/YitFc mice shown in this study have some common features with those of CD patients and suggest further that this mouse model is suitable for analyzing the effects of dysbiosis on pathophysiology of CD. In addition, SAMP1/YitFc mice show the improvement of the disease state by administration of TNFα antibody used for clinical treatment and other conventional therapies (58). Taken together, it is suggested that SAMP1/YitFc mouse is a suitable preclinical model for analyzing the pathogenesis and pathophysiology of CD. Although failure of homeostasis of both the immune system and epithelial cells in the intestine via dysbiosis has been considered to contribute to the pathophysiology of CD (18), the precise relationship remains unclear. Paneth cells, one of the major small intestinal epithelial lineages, and the α-defensins they secrete into the intestinal lumen are known to regulate the composition of the intestinal microbiota (12). Also, expression levels of HD5, a human Paneth cell α-defensin, have been reported to decrease (59) or elevate (60) in the ileum of CD patients, so that the relationship between CD and the amount of HD5 remain controversial. Furthermore, abnormal localization of secretory granules containing α-defensins in Paneth cells of CD patients has been reported in association with progression of dysbiosis and an increase of relapse rates (31, 61). These studies suggest that Paneth cell

dysfunction accompanied by alterations in α-defensin activities cause dysbiosis and further relates to disease progression of CD.

Therefore, we focused on relationships between Paneth cells and disease onset and progression of ileitis in SAMP1/YitFc mice in this study. We showed that aberrant cellular localization of abnormal Paneth cells increases along with disease progression, consistent with previous reports that intermediate cells with immature granules appear and increase in crypts and on lower regions of villi in SAMP1/YitFc mice (40). We further revealed that ER stress occurs in abnormal SAMP1/YitFc mouse Paneth cells, consistent with reported relationships between CD pathology and Paneth cell ER stress (25). Genes involved in resolving ER stress, for example, UPR-related XBP1 (26) and AGR2 (27), autophagy-related ATG16L1 (28), IRGM1 (29), and LRRK2 (30) are known susceptibility genes for CD, and deletions or mutations in these genes lead to Paneth cell abnormalities. We showed that genes regulated downstream of XBP1 are modulated as disease progresses in SAMP1/YitFc mice. For example, levels of two ER stress makers, GRP78 and calreticulin, increase, and levels of MIST1, which inhibits ER stress (42), decrease, suggesting that pathophysiology in SAMP1/YitFc mice shares certain pathways with CD patients. In addition, environmental risk factors of CD such as high-fat diet (23) and zinc deficiency (62) induce ER stress in Paneth cells (63, 64). Vitamin D deficiency, a CD risk factor, induces defective autophagy in mouse Paneth cells (57), suggesting that diverse environmental factors may induce ER stress in Paneth cells. Moreover, CD patients that have Paneth cell ER stress due to ATG16L1 gene mutations are colonized by enteroinvasive E. coli which induce enteritis in higher rates (32). Thus, the relationship between ER stress in Paneth cells and dysbiosis in CD has been suggested, although the mechanisms that link ER stress and dysbiosis remain unclear. ER stress induces misfolding of disulfide bonds during protein synthesis in cells (33, 34). α-Defensins secreted from Paneth cells are characterized by three intramolecular disulfide bonds in their tertiary structures (65, 66) and reduced-form HD5 with no disulfide bonds has been detected in Paneth cells of CD patients (36). Thus, we hypothesized that the ER stress that occurs in abnormal SAMP1/YitFc mouse Paneth cells may result in misfolding of Crps. In testing this hypothesis, we found that rCrps induced by ER stress were secreted from SAMP1/YitFc mouse Paneth cells into the intestinal lumen, indicating that ER stress in Paneth cells causes misfolding of α-defensins, regulators of the intestinal microbiota, resulting in secretion of rCrps.

Previously, we reported that oxCrps elicit no or minimal bactericidal activity against eight species of commensal bacteria, including Bifidobacterium bifidum and Lactobacillus casei, whereas rCrps kill these commensals (35). Therefore, we analyzed the relationship between dysbiosis and secretion of rCrps into the intestinal lumen in this study. rCrps in feces positively correlated with dysbiosis, and rCrp1 elicited significantly greater potency than oxCrp1 against A. colihominis in in vitro bactericidal assays, which decreases in SAMP1/YitFc mice in vivo. These findings suggest that secretion of rCrps leads to dysbiosis and contributes to progression of ileitis. Although the factors that determine the differential bactericidal spectra of ox and rCrps remain unclear, selective permeabilization of bacterial cell membranes by ox and rCrps is a possibility (35, 67). In part, this notion is supported by the findings that Crp4 variants that do not form disulfide bonds accumulate at bacterial membrane surfaces, whereas oxCrp4 binds to bacterial membranes and translocates into them (67). Also, rCrp4 may cause stronger membrane depolarization than oxCrp4 peptides (35). In this study, we have also shown that rCrps secreted into the small

intestinal lumen by Paneth cells reach the colonic lumen and can be recovered in feces. Reduced-form α-defensins are sensitive to degradation in vitro by proteinases which also are abundant in the intestinal lumen (45). Nevertheless, we recovered intact rCrps from feces, suggesting that mechanisms exist for protection from luminal proteinases. Perhaps, as has been reported for reduced-form HD5, proteolytic stability may result by forming complexes with $Zn^{2+}$ in vitro (68). Also, fecal concentrations of the serine protease inhibitor $α_1$-antitrypsin increase in CD patients compared with healthy subjects (69). Perhaps, these and additional mechanisms protect reduced-form α-defensins from degradation in the intestinal lumen. We speculate that in CD patients, such factors allow reduced-form α-defensins to persist and induce dysbiosis.

This study introduces a new concept into the onset of CD in which ER stress in Paneth cells results in the secretion of reduced-form α-defensins into the intestinal lumen, and that dysregulated spectrum of bactericidal activities of reduced-form α-defensins induce dysbiosis. Differences in α-defensin bactericidal activities due to the presence or absence of disulfide bonds have been reported in mouse and human. For example, reduced-form HD5 has lower bactericidal activities against *E. coli* and *Staphylococcus aureus* compared with oxidized-form HD5 (70). Reduced-form HD6 inhibits the growth of certain commensal bacteria including *Bifidobacterium adolescentis*, whereas oxidized-form HD6 has no such effect (71). Moreover, reduced-form HD5 has been detected in human ileal tissue (36) and also in the intestinal lumen (72). These studies support the view that dysbiosis due to α-defensin misfolding may occur not only in a mouse model of CD but also in patients. Additional clinical studies are needed to test this possibility. Furthermore, ER stress in Paneth cells may be caused by varied environmental factors in addition to genetic risk factors, and Paneth cell defects accompanied by ER stress are associated with diseases related with dysbiosis such as obesity (73), ischemia/reperfusion (74), and alcoholic liver disease (75). Possibly, α-defensin misfolding and the secretion of reduced-form α-defensins caused by Paneth cell ER stress may also be associated with onset and progression of diseases. In the future, comprehensive analyses including tertiary structure of α-defensins in the intestinal lumen, composition of the intestinal microbiota, and pathophysiology of dysbiosis-related diseases may identify new pathogenetic mechanisms triggered by α-defensin misfolding. Such findings should contribute further to the development of novel diagnostic methods and therapeutics that target reduced-form α-defensins.

# Materials and Methods

### Mice

SAMP1/YitFc mice were purchased from Charles River Laboratories Japan, Inc. and propagated at Hokkaido University. ICR mice were purchased from CLEA Japan, Inc. at 3 wk. All mice were housed under conventional conditions maintained under a 12-h light/dark cycle with water and food provided ad libitum. All animal experiments in this study were conducted after obtaining approval from the Institutional Animal Care and Use Committee of the National University Corporation at Hokkaido University in accordance with Hokkaido University Regulations of Animal Experimentation.

### Histological analysis

Mice were euthanized by isoflurane inhalation, then ileum (distal one-third of the small intestine) was opened, debris rinsed with ice-cold PBS (-), and rolled longitudinally into a Swiss-roll configuration. Ileal tissues were fixed in 10% buffered formalin, embedded in paraffin, and sliced into 4-μm sections. After deparaffinization, sections were stained with HE and Alcian blue. For histological evaluation, 1 through 10 well-orientated crypt-villus axes indicating complete longitudinal sectioning were selected from each HE staining section in a sequential order from the distal end. Each crypt-villus axis was evaluated following three categories: (a) inflammatory infiltration (the number of inflammatory cells in the lamina propria), (b) villus distortion (ratio of villus length to crypt depth), and (c) thickening of muscle layer (thickness of muscle layer directly under the crypt). Each crypt-villus axis was graded from 0 to 3 in each category based on the criteria shown in Table S1. To determine the range for scoring in (a)–(c), all sections obtained from each ICR and SAMP1/YitFc mouse were pre-examined. In (a) and (c), mean ± 2 SD of measured values in ICR mice was set as the range of score = 0. Then, the range between mean + 2 SD in ICR mice and maximum value in SAMP1/YitFc mice was divided into three equal parts, and each part was set as the range of score 1, 2, and 3. In (b), mean ± 0.5 of measured values in ICR mice was set as the range of score = 0. Then, the range between mean −0.5 in ICR mice and minimum value in SAMP1/YitFc mice was divided into three equal parts, and each part was set as the range of score 1, 2, and 3. Under these criteria, more than 90% of ICR mice were scored to 0. In (d), the criteria were determined based on a previous study (76). Scores of (a)–(c) for each mouse were calculated as averages of crypt-villus axes evaluated in the sections. Scores of (d) crypt abscess in each mouse were calculated by multiplying 0.5 and the number of crypt abscesses in the entire area of the sections together. Total inflammatory score of each mouse was calculated by a summation of (a)–(d).

### Immunohistochemistry

Ileal sections were deparaffinized, rehydrated, and boiled in antigen retrieval solution (pH 9.0) (Nichirei Bioscience) at 105°C for 20 min. Sections were blocked in 20% Block Ace (Dainippon Pharmaceutical) in PBS containing 5% goat serum (Sigma-Aldrich) at room temperature for 30 min. After blocking, the sections were incubated with primary antibodies: anti-Muc2 (1 μg/ml, sc-15334; Santa Cruz Biotechnology), anti-Crp1 (1 μg/ml, 77-R63, self-produced), anti-GRP78 (1 μg/ml, ab21685; Abcam), anti-calreticulin (10 μg/ml, #62304; Cell Signaling Technology), anti–Ephrin-B2 (10 μg/ml, AF496; R&D systems), and anti-MIST-1 (0.25 μg/ml, ab187978; Abcam) at 4°C for overnight. Then, the sections were incubated with fluorescent-conjugated secondary antibodies (Life Technologies) at room temperature for 1 h. After incubation, the sections were covered with coverslips mounted with VECTORSHIELD medium with DAPI (Vector Laboratories), sealed with nail polish and dried. Fluorescence images were observed using LSM510 Confocal Laser Scanning Microscope (Carl Zeiss).

### TEM

5-mm-long segments of terminal ileum were fixed with 2% parafor-maldehyde and 2% glutaraldehyde at 4°C overnight. After fixation, the samples were post-fixed with 2% osmium tetroxide at 4°C for 2 h. The samples were dehydrated and embedded in Quetol-812 epoxy resin (Nisshin EM). Then, ultrathin sections with a thickness of 70 nm were cut using Ultracut UCT (Leica Microsystems), mounted on copper grids, and stained with 2% uranyl acetate at room temperature for 15 min, and then with lead stain solution (Sigma-Aldrich). The sections were imaged with a JEM-1400Plus transmission electron microscope (JEOL Ltd.) at an acceleration voltage of 100 kV. For quantitative analysis of secretory granule numbers and area, all granules in randomly selected Paneth cells were measured. For quantitative analysis of ER lumen diameter, five ER cisternae were selected in a sequential order from the closest to nuclei of each selected Paneth cell, and then the length of most expanded part of each ER cisterna was measured and averaged as ER lumen diameter of each Paneth cell. All measurements were conducted by using ImageJ software (National Institutes of Health, http://rsb.info.nih.gov/ij/).

### Crypt isolation

Crypts were isolated and fractionated from 5 cm of terminal ileal tissues as previously described (77). The fraction with >70% purity of crypts was used in this study.

### SDS–PAGE Western blot

Isolated ileal crypts were lysed in radioimmunoprecipitation assay (RIPA) buffer (50 mM Tris–HCl, pH 8.0, 150 mM NaCl, 0.1% Triton-X 100, and 0.1% SDS) for protein extraction, and protein content of the sample was measured by using the BSA Protein Assay Kit (Thermo Fisher Scientific) following the manufacturer's protocol. 20 μg of protein extracted from each crypt sample was dissolved in tricine sample buffer (Bio-Rad) and analyzed by SDS–PAGE using Mini-PROTEAN TGX Precast gels (Bio-Rad) and transferred to 0.22 μm polyvinylidene fluoride (PVDF) membrane at 1.3 V for 7 min using Trans-Blot Turbo Transfer system (Bio-Rad). After transfer, the membranes were blocked by 5% BSA (Sigma-Aldrich) in 0.1% TBS-T at 4°C for overnight and then reacted with the following primary antibodies: anti-ATF4 (1/1,000, #11815; Cell Signaling Technology), anti-phospho-IRE1α (1 μg/ml, NB100-2323; Novus Biologicals), anti-total IRE1α (1 μg/ml, NB100-2324; Novus Biologicals), anti-ATF6 (1 μg/ml, NBP1-40256; Novus Biologicals), anti-GRP78 (1/1,000, #3177; Cell Signaling Technology), and anti-HPRT1 (0.018 μg/ml, ab109021; Abcam) at 4°C overnight. After washing with 0.1% TBS-T, the membranes were incubated with HRP-conjugated secondary antibodies (GE Healthcare) at room temperature for 1 h. After incubation, the blots were visualized by Chemi-Lumi One Ultra (Nacalai Tesque). For the quantification of each molecule, band intensities were determined by ImageJ software.

### Protein purification from small intestinal tissue

Full length of small intestinal tissue was obtained from 10 ICR and 10 SAMP1/YitFc mice and rinsed free of debris with ice-cold PBS (-).

Tissue samples were minced in 100 mM iodoacetamide (Nacalai Tesque) in ice-cold PBS (-) containing Complete Mini protease inhibitor cocktail (Roche Applied Science). Then, the tissues were homogenized using a Potter homogenizer and rotor–stator homogenizer TissueRuptor (QIAGEN). For alkylation reactions, the samples were incubated at room temperature for 1 h. The samples were diluted to a final concentration of 30% (vol/vol) by acetic acid (Nacalai Tesque) and rotated at 4°C for overnight. Precipitates were removed by centrifugation (15,000$g$ for 30 min) at 4°C. Supernatants were dialyzed using Spectra/Por7 membranes (Spectrum Labs) against 5% (vol/vol) acetic acid at 4°C and then lyophilized. Lyophilized extracts were resuspended in 10 ml of 5% (vol/vol) acetic acid and subjected to purification using preparative acid–native PAGE using Model 491 Prep Cell device (Bio-Rad). Resuspended extracts were applied to the apparatus filled with stacking gel (10% T, 3% C) and separating gel (16% T, 3% C), and then electrophoresed at 200 V. Eluents were collected from the anodal reservoir with 5% (vol/vol) acetic acid at a flow rate of 1 ml/min and fractionated every 10 min. To determine whether proteins are dissolved in the collected fractions, 1/20 volume of each fraction was electrophoresed on AU-polyacrylamide gel (12.5% T, 3% C) at 150 V (78). After electrophoresis, proteins were visualized by staining with Coomassie Blue R-250.

### AU-PAGE Western blot

Samples (1/20 vol of fractions containing proteins extracted from the small intestinal tissue) were resolved by AU-PAGE at 150 V and transferred to 0.1-μm nitrocellulose membrane (GE Healthcare) using a semidry apparatus (ADVANTEC) at 2.5 mA/cm$^2$ for 30 min. Chemically synthesized ox and rCrp1 prepared as described (44) were used as standards. The membranes were blocked by BSA-free StabilGuard Immunoassay Stabilizer (Surmodics) at room temperature for 1 h, incubated with biotinylated anti-Crp1 antibody (1 μg/ml, 76-R29, self-produced), which reacts to both ox and rCrp1–4 and 6 (Fig S2) at 4°C overnight. Membranes were incubated with HRP-conjugated streptavidin (GE Healthcare) at room temperature for 1 h. After incubation, the blots were visualized by Chemi-Lumi One Ultra.

### Dot blot analysis

50 ng of chemically synthesized Crps were dissolved in 5% acetic acid and pipetted onto a 0.1-μm nitrocellulose membrane (GE Healthcare). The membrane was dried and treated as described for AU-PAGE Western blot.

### Enteroid culture and Paneth cell live imaging

Enteroid culture and Paneth cell live imaging have been described (77, 79). Briefly, crypts were isolated from the proximal half of small intestines of 18–20-wk ICR or SAMP1/YitFc mice and cultured for 3 d. Cultured enteroids were transferred onto collagen-coated glass bottom eight-well chamber cover slips (Matsunami) at 100 enteroids/well and differential interference contrast images of Paneth cells in enteroids were acquired by confocal microscopy (A1; Nikon) before and 10 min after adding 10 μM CCh to the culture

medium (Sigma-Aldrich). To quantify Paneth cell granule secretion, Paneth cell granule area were measured pre- and post-stimulation by using NIS-Elements AR and percent area granule secretion were calculated as before ([77]). For Paneth cell survival analysis, enteroids were exposed to 10 $\mu$M CellEvent caspase-3/7 Green Detection Reagent (Thermo Fisher Scientific) in the culture medium and time-lapse images of Paneth cells stimulated by 10 $\mu$M CCh were acquired at 15 frames/sec at low laser power. 27 ICR mouse Paneth cells and 24 SAMP1/YitFc mouse Paneth cells were analyzed.

### Quantification of fecal cryptdin

Fecal samples collected from each mouse were dried by lyophilization. After lyophilization, fecal samples were pulverized to powder using a bead beater-type homogenizer lT-12 (TAITEC). 10 mg of fecal powder was suspended with 100 $\mu$l of 30% (vol/vol) acetic acid and vortexed at 4°C overnight. Precipitates were removed by centrifugation (15,000$g$ for 30 min) at 4°C. Supernatants were collected and concentrated in a SpeedVac concentrator (Thermo Savant) and resuspended in 20 $\mu$l of PBS (-). Trypsin (Thermo Fischer Scientific) was dissolved in PBS (-) and adjusted to 20 ng/$\mu$l. 1 $\mu$L of fecal extract and 9 $\mu$l of trypsin solution were mixed and incubated at room temperature for 1 h. Tryptic digests of fecal samples were resolved by Tris-Tricine SDS–PAGE ([80]) at 30 mA/gel and transferred to 0.22-$\mu$m PVDF membranes at 1.3 V for 7 min using Trans-Blot Turbo Transfer system. After the transfer, the membranes were treated as described above. For quantification of Crps, band intensities were determined by ImageJ software.

### DNA purification

Fresh fecal samples collected from mice, were immediately snap-frozen, and stored at –80°C. Total DNA was extracted from 200 mg fecal samples using QIAamp Fast DNA Stool Mini Kit (QIAGEN) following the manufacturer's protocol. Final DNA concentrations were determined at 260 nm using a NanoDrop 2000 spectrometer (Thermo Fischer Scientific).

### 16S rDNA sequencing

16S ribosomal RNA genes were amplified by PCR from each fecal DNA sample using universal primer set of Bakt 341F (5-cctacgggnggcwgcag) and Bakt 805R (5-gactachvgggtatctaatcc) which covers the V3-V4 variable region ([81]). PCR amplification was performed in 25-$\mu$l-volume reaction mixtures containing 12.5 ng of template DNA, 200 nM of each primer, and 1× KAPA HiFi Hot Start Ready Mix (Kapa Biosystems) under the following conditions: 95°C for 3 min, 25 cycles of 95°C for 30 s, 55°C for 30 s, and 72°C for 30 s, followed by 72°C for 5 min. PCR products were purified with AMPure XP beads (Beckman Coulter). After purification, sequencing adapters containing sample-specific 8-bp barcodes were added to the 3′- and 5′- ends by PCR using the Nextera XT Index Kit v2 Set B (Illumina) in 50 $\mu$l of reaction mixtures containing 5 $\mu$l of PCR amplicon, 5 $\mu$l of each indexing primer and 1× KAPA HiFi Hot Start Ready Mix under the following conditions: 95°C for 3 min, eight cycles of 95°C for 30 s, 55°C for 30 s, and 72°C for 30 s, followed by

72°C for 5 min. Each amplicon was purified, quantified using the Qubit dsDNA HS Assay Kit (Invitrogen), and then adjusted to 4 nM. Amplicons were pooled 4 $\mu$l and subjected to quantification using KAPA Library Quantification Kit LightCycler 480 qPCR Mix (Kapa Biosystems) and then diluted to 4 pM. The amplicon library was combined with 5% equimolar PhiX Control v3 (Illumina) and sequenced on a MiSeq instrument using the MiSeq 600-cycle v3 kit (Illumina).

### 16S rDNA–based taxonomic analysis

Illumina pair-end reads FASTQ files were obtained after 16S rDNA sequence, and the OTU classification and diversity analyses were performed using QIIME version 1.9.1 ([82]) according to a previously described method (http://doi.org/10.5281/zenodo.1439555). Briefly, adapter sequences and low-quality bases were trimmed from sequenced reads by BBDuk, and pair-end reads were merged by BBmerge. Both BBDuk and BBmerge are included in the BBtools software suite (http://jgi.doe.gov/data-and-tools/bbtools/). Chimeric sequences were removed by UCHIME ([83]). The OTUs were classified taxonomically into five rank categories (phylum, order, class, family, and genus) by using the SILVA 12_8 reference database based on a 97% similarity threshold with UCLUST ([84]). Using Qiime workflow, $\alpha$-diversity was estimated by observed OTUs and PD whole-tree and $\beta$-diversity was estimated by unweighted UniFrac distance and visualized with Principal Coordinate Analysis. Statistical significance of $\beta$-diversity was determined by PERMANOVA in Qiime.

### Bactericidal assays

*A. colihominis* JCM 15631 was purchased from the Institute of Physical and Chemical Research (RIKEN). The bacterium was cultured in Eggerth–Gagnon liquid medium supplemented with defibrinated horse blood under anaerobic conditions using the Anaero Pack system (Mitsubishi Gas Chemical) at 37°C. Exponential-phase bacteria were centrifuged at 3,000$g$ for 5 min at 4°C, washed twice, and then resuspended in PBS (-) diluted 1:1 with Milli-Q water. The OD$_{600}$ was measured to determine bacterial cell numbers. 50 $\mu$L of suspension containing 1,000 CFU per aliquot were mixed with equal volumes of ox or rCrp1 to final concentrations from 0 to 1.35 $\mu$M, incubated under anaerobic conditions at 37°C for 1 h, and mixtures were plated on Eggerth–Gagnon agar plates and incubated anaerobically at 37°C. Bacterial survival rates at each concentration were calculated from the number of surviving colonies relative to peptide-unexposed controls.

### Antibiotic treatment

SAMP1/YitFc mice were subjected to antibiotic (Abx) treatments for 6 wk from 4 to 10 wk as described ([85]). Briefly, 1 g/l ampicillin sodium salt (Sigma-Aldrich) dissolved in distilled water was administered ad libitum in drinking water, and 10 ml/kg body weight of an antibiotic cocktail consisting of 5 mg/ml vancomycin hydrochloride (Wako pure chemical industries), 5 mg/ml neomycin trisulfate salt hydrate (Sigma-Aldrich), and 10 mg/ml metronidazole (Sigma-Aldrich) dissolved in distilled water was orally administered

every 12 h via silicon sonde (Fuchigami Kikai) to Abx-treated mice (Fig S12A). Ampicillin-supplemented drinking water was renewed every 7 d. Fresh Abx cocktail was mixed every day. As a control, distilled water containing no antibiotics was administered ad libitum and orally administered via silicon sonde to water-treated mouse. Fresh fecal samples from each mouse were collected at 4 and 10 wk. Total DNA was purified from fecal samples and subjected to PCR amplification as described for 16S rDNA sequencing. To test for the persistence of the intestinal microbiota, PCR products from each fecal sample were electrophoresed in agarose gels and visualized by ethidium bromide staining.

### Rectal administration of reduced-form Crp1

5-wk ICR mice were habituated to the experimental condition for 7 d. After the acclimation period, fresh fecal samples for microbiota analyses were collected at day 0, day 2, and day 4. At day 1 and day 3, single doses of 200 μl of 1 mg/ml rCrp1 dissolved in distilled water or 200 μl of distilled water were administered rectally to rCrp1-treated mice or control mice, respectively (Fig S13A). Substances were deposited to a depth of 2 cm by using stainless steel sonde.

### Statistical analyses

All statistical analyses were performed using the Prism ver. 7.0 software (GraphPad). All tests were performed as two-sided. Statistical significance was determined by $t$ test or Mann–Whitney's U test between two groups, one-way ANOVA followed by Tukey's post hoc test among more than three groups, and two-way ANOVA followed by Bonferroni post hoc test for repeated measurements. Correlation analysis was performed by Pearson's correlation coefficients test. $P < 0.05$ was considered statistically significant.

# Data Availability

Raw sequencing files for the 16S rDNA sequencing data sets are available at Sequence Read Archive (PRJNA622264, PRJNA622384).

# Supplementary Information

# Acknowledgements

We are grateful to Prof AJ Ouellette (University of Southern California) for helpful discussions. We thank Ms Aiko Kuroishi and Ms Mutsuko Tanaka for experimental support. This study was supported by grants from the Japan Society for the Promotion of Science KAKENHI Grant Number 17K11661 to K Nakamura and 18H02788 to T Ayabe and the Center of Innovation Program from Japan Science and Technology Agency Grant Number JPMJCE 1301 to K Nakamura and T Ayabe.

## Author Contributions

Y Shimizu: conceptualization, resources, data curation, formal analysis, validation, investigation, visualization, methodology, and writing—original draft.
K Nakamura: conceptualization, resources, data curation, formal analysis, supervision, funding acquisition, validation, methodology, and writing—review and editing.
A Yoshii: formal analysis, investigation, and methodology.
Y Yokoi: formal analysis, investigation, visualization, and methodology.
M Kikuchi: formal analysis and investigation.
R Shinozaki: formal analysis and investigation.
S Nakamura: formal analysis and investigation.
S Ohira: formal analysis and investigation.
R Sugimoto: formal analysis and investigation.
T Ayabe: conceptualization, supervision, funding acquisition, project administration, and writing—review and editing.

## Conflict of Interest Statement

The authors declare that they have no conflict of interest.

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
