## [Reviewer comments · Life Science Alliance]

Life Science Alliance

Paneth cell α -defensin misfolding correlates with dysbiosis and ileitis in Crohn's disease model

Yu Shimizu, Kiminori Nakamura, Aki Yoshii, Yuki Yokoi, Mani Kikuchi, Ryuga Shinozaki, Shunta Nakamura, Shuya Ohira, Rina Sugimoto, and Tokiyoshi Ayabe

DOI: <https://doi.org/10.26508/lsa.201900592>

Corresponding author(s): Tokiyoshi Ayabe, Hokkaido Univ

Review Timeline:

Submission Date:	2019-10-29
Editorial Decision:	2019-11-02
Appeal Requested:	2019-11-04
Editorial Decision:	2019-11-06
Revision Received:	2019-11-06
Editorial Decision:	2019-12-08
Revision Received:	2020-03-19
Editorial Decision:	2020-03-26
Revision Received:	2020-04-07
Accepted:	2020-04-07

Scientific Editor: Andrea Leibfried

Transaction Report:

November 2, 2019

Re: Life Science Alliance manuscript #LSA-2019-00592-T

Prof. Tokiyoshi Ayabe
Hokkaido Univ
Faculty of Advanced Life Science
Sapporo 060-0810
Japan

Dear Dr. Ayabe,

Thank you for submitting your manuscript entitled "Paneth cell α -defensin misfolding correlates with dysbiosis and ileitis in Crohn's disease model mice" to Life Science Alliance. We have now assessed your paper and discussed it within our editorial team.

Your analyses show that in a murine model for Crohn's Disease, Paneth cells show signs of ER stress and produce less α -defensins. Reduced α -defensin production/secretion correlates with alterations of the gut microbiome.

We appreciate that your data suggest that Paneth cell dysfunction drives intestinal inflammation in mice. However, upon discussing your data within our editorial team, we concluded that the value provided by the observations remains too limited at this stage given the known links between inflammatory bowel diseases and dysbiosis and between α -defensin release from Paneth cells and microbiota composition. We think that a more causal link between Paneth cells/ α -defensin and the disease would be expected by the community at this stage. We have thus decided not to subject your manuscript to a lengthy external review process.

I am sorry that our answer on this occasion is not more positive, and I hope that this outcome will not dissuade you from submitting other manuscripts to us in the future.

Thank you for your interest in Life Science Alliance.

With kind regards,

Andrea Leibfried, PhD
Executive Editor
Life Science Alliance

Thank you for reviewing our manuscript #LSA-2019-00592-T. I just received your decision not to subject to external review process. However, you might be seriously misunderstanding our findings. Now, I would like to have a chance to explain details of the findings in our manuscript, so that you would rethink the decision at this time.

In this manuscript, we clarified ER stress occurring in Paneth cells results in misfolded alpha-defensin production and secretion of reduced form of alpha-defensin, which kills both pathogenic and symbiotic microbes. Normally, only oxidized form of alpha-defensin is secreted from Paneth cells and no reduced form of alpha-defensin exist in the intestinal lumen. We found that the secreted reduced form of alpha-defensin, disrupts the gut ecosystem, inducing dysbiosis and resulting in severe ileitis. Our findings add completely new mechanism and previously unknown important insights into the relationship between Paneth cell biology and dysbiosis resulting in disease.

Thank you for giving me this opportunity to respond to your email. Thank you very much for your kind reconsideration.

I am looking forward to hearing from you.

MS: LSA-2019-00592-T

Prof. Tokiyoshi Ayabe
Hokkaido Univ
Faculty of Advanced Life Science
Sapporo 060-0810
Japan

Dear Dr. Ayabe,

Your manuscript entitled "Paneth cell α -defensin misfolding correlates with dysbiosis and ileitis in Crohn's disease model mice" has now been reconsidered, and I am pleased to let you know that we have decided to send your manuscript for external review.

We will let you know when the reviews have been received and a decision has been made.

Yours sincerely,

December 8, 2019

Re: Life Science Alliance manuscript #LSA-2019-00592-TR-A

Prof. Tokiyoshi Ayabe
Hokkaido Univ
Faculty of Advanced Life Science
Sapporo 060-0810
Japan

Dear Dr. Ayabe,

Thank you for submitting your manuscript entitled "Paneth cell α -defensin misfolding correlates with dysbiosis and ileitis in Crohn's disease model" to Life Science Alliance. The manuscript was assessed by expert reviewers, whose comments are appended to this letter.

As you will see, the reviewers point out various weaknesses that would need to get addressed in revision. Should you be able to do so, we would be happy to consider a revised version further. But please consider your options carefully as a lot of effort would be required to add the insight requested by reviewer #2 and #3 and in order to render your manuscript valuable to the community. Further, reviewer #3 points out that Crohn's disease patients actually have less defensin expression, questioning the relevance of the model you want to put forward as a relevant one for Crohn's disease studies. This concern should get addressed in a constructive way, too. Should you prefer to submit your manuscript elsewhere given the revision requests, please let us know.

The typical timeframe for revisions is three months, but this can get extended. Please note that papers are generally considered through only one revision cycle, so strong support from the referees on the revised version is needed for acceptance.

Thank you for this interesting contribution to Life Science Alliance. We are looking forward to receiving your revised manuscript.

Sincerely,

B. MANUSCRIPT ORGANIZATION AND FORMATTING:

Reviewer #1 (Comments to the Authors (Required)):

Dear Editor,

The paper by Shimizu et al., entitled "Paneth cells alpha-defensin misfolding correlates with dysbiosis and ileitis in Crohn's disease model mice" is of interest in the field of gastrointestinal chronic disease as it shows important data about the role of cryptidins in the control of microbiota composition and inflammation.

The authors used SAMP1/YitFc mice developing ileitis to analyze ER stress and cryptidins release by Paneth cells. The authors show clear correlations between inflammatory score and eosinophilic granules, changes in oxidization of Crp during the course of inflammation leading to changes in microbiota composition in their mouse model. This paper mostly shows correlations, no mechanistic data are presented. However, these correlations seem to be of interest in the context of IBD research.

Here are minor suggestions which could improve the paper:

Figure 1a: Please add arrows to show Paneth cells on the villi.

Figure 1f: The authors should split the channels to better show the expression of MUC2 and Crp1.

Figure 2: The western-blots are not very convincing. It is difficult to determine which band have been quantified (Especially for pIRE1 and GRP78). The quantifications (fig 2d) do not show standard errors bars.

Figure 3: What do "fraction" stands for? Does that mean different part of the small intestine?

Please precise. Please also precise the number of experiment performed in the legend.

Supplementary figure 1: Please label the westerns/dot blots. It is not possible to understand what is shown on the figures as currently presented

line 60: "suggests show"

line 215: On the graphs, it appears that Bacteroides was positively correlated to inflammatory score and Anaerotruncus was negatively correlated to inflammatory score. The text should be corrected accordingly.

line 230-232: Please soften the conclusion: "could be involved" instead of "is involved".

Line 234-236: The sentence is not clear.

The English language should be checked by a native English speaker as many errors remain in the text.

Reviewer #2 (Comments to the Authors (Required)):

Using a mouse model of Crohn's disease, the authors find evidence of ER-stress-induced disruption of in vivo Paneth Cell function. A model is proposed whereby altered protein folding in the ER results in increased secretion of reduced forms of cryptidins, which selectively alters the microbiome in favour of more pathogenic bacterial strains. This dysbiosis is considered a key factor in the progression of Crohn's disease in humans and may reveal Paneth cells as novel therapeutic targets.

Although much of the data is correlative rather than being strictly mechanistic, the study does provide solid evidence for an association between dysfunctional Cryptidin secretion from Paneth cells and onset of Crohn's disease.

Critique

- 1) Authors claim significant crypt elongation during progression of ileitis in their mouse model - please quantify this. The specified crypt abscess is also not clear from Figure 1.
- 2) Reduction in the size of Paneth cell granules is clear - does the number of granules per cell also change during disease progression? Do the dysfunctional Paneth cells still express lysozyme?
- 3) Presence of altered Paneth cells in upper crypts and villi not clear from figure 1. What causes

this? Dysregulated Wnt signalling causing loss of EpH/ephrin boundaries?

4) Are the defective Paneth cells found throughout the intestines? If so, why does disease typically present in the terminal ileum? Is the Paneth cell phenotype seen in other mouse models of Crohn's disease (eg, TNFalpha-overexpressing mice)?

5) Co-expression of Muc2 and Crps appears restricted to the upper crypts/villi rather than the Paneth cell zone at the crypt base - this suggest that goblet cells might be converting into Paneth cells rather than vice versa - could these altered goblet cells be contributing to disease progression?

6) Figure 2c - Western blots are not publication quality - It is very difficult to concur with the claims of altered ATF4/IRE expression based on these data. Why not use ER stress markers in IHC/IF? What about IRE1B and downstream targets of Xbp1 (via q-PCR)? It is also questionable whether the use of whole crypt extracts allows conclusions regarding ER stress specifically in Paneth Cells - multiple cell types in the crypts may be undergoing ER stress. Why not purify the Paneth cells from the normal and diseases intestines (for example using the CD24^{hi}/SSChi protocol from Sato et al) for more specific analyses of the changes to Paneth cells? This would also facilitate a more unbiased approach to evaluating changes in the Paneth cells during disease progression. Sorting Paneth Cells from intestinal organoids derived from human Crohn's disease patients would also facilitate validation of the major findings and establish the relevance for human disease.

7) Are misfolded forms of Defensins present in human fecal samples from Crohn's disease patients?

8) Discussion needs to be much more concise and presenting detailed discussion rather than an extended re-iteration of the findings.

Reviewer #3 (Comments to the Authors (Required)):

The manuscript by Shimizu et. al. attempts to provide a link between SAMP1/YitFc mice, a spontaneous ileitis mouse model and its ileitis development and Paneth cell abnormality. The authors states that Paneth cell abnormality is the cause of dysbiosis and ileitis development. The work is built on the previous knowledge of this mouse model. It provides incremental knowledge to our understanding of the model, and the data are descriptive in nature.

Major concerns:

1. Crohn's disease (CD) patients are known to have reduced defensin expression in Paneth cells (PMID: 19904243), and yet in this study, SAMP1/YitFc mice have increased Paneth cells and cryptidin expression. The discrepancy calls into the question the relevance as a "CD" mouse model.
2. While Paneth cell defects can lead to dysbiosis, an alternative hypothesis is that the microbes trigger Paneth cell abnormality. Examples of how environmental factors can affect Paneth cells include PMID: 20602997, 30137026. To exclude that dysbiosis could lead to Paneth cell defects, an experiment can be set up where mice are given antibiotics to deplete these microbes, and determine if Paneth cells remain abnormal. Alternatively, if the authors have these mice bred in gnotobiotic facility, that will also provide tremendous insight.
3. Along the same line, to claim that the excessive reduced-form cryptidin alters microbiota composition, the authors need to show that supplement with reduced-form cryptidins can alter the microbiota in wild type mice.
4. The authors' claim that altered cryptidins are being "secreted" into the intestinal lumen lacks experimental proof. The Paneth cells may simply die of excessive ER stress and as a result, the cryptidins are being released into the lumen. There is no data to define Paneth cell survival, turnover, cell death, nor a proper in vivo secretion assay (one such example would be PMID:

18849966).

5. It is known that the genetics of the SAMP1/YitFc mice is unclear. As a result, the use of ICR mice (with no description of background...etc) as a control for microbiome studies is suboptimal. What is the baseline difference of microbiota compositions between SAMP1/YitFc and ICR mice? This data should be included. The authors should also include PCoA plot analyses for 0, 4, and 20 weeks time points.

6. Many key data lack quantification: For example, how often did the authors observe ER changes in TEM in Figure 2? From previous study (PMID: 15793286), this may be a very small proportion.

Minor comments:

1. The authors should provide rationale for testing cryptidin 4 but rather than the more Paneth cell-specific cryptidin 4.

2. Figure 1: this is essentially confirmatory of what has been previously published (PMID: 15793286), yet in figure 1f there seems to be very few overlap staining between cyp1 and muc2.

3. Likewise, it was not clear from Fig.1a how the Paneth cell granule sizes are different between the two strains of mice. The authors need to provide methodology how they measured the granule size (how many granules, how many Paneth cells were counted), as well as a blown-up picture to highlight the size difference.

4. In Fig. 2. the authors only showed 2 mice each for each of the panel. This is insufficient for statistics.

5. The use of the term "reduced cryptidins" may be confusing for some (as compared to reduced quantity of cryptidins). Suggest to change it (such as reduced-form cryptidins) to avoid such confusion.

6. Figure 5, legend: there does not seem to have ICR mice data in this figure.

7. Supplementary figure 1 is incomprehensive as there was no label to describe these blots.

We greatly appreciate the editor's and the referees' constructive comments and suggestions for improvement of our paper. In this revised manuscript, we have addressed all points that were raised by the referees. Our point-by-point response letters to each referee are shown below.

Response to Referee #1

We thank the referee for valuable suggestions and comments. Based on those comments, we have corrected and updated the revised manuscript as described in the following point-by-point responses.

1-1: Figure 1a: Please add arrows to show Paneth cells on the villi.

Accordingly, arrows showing the abnormal Paneth cells in crypts at 10 w and on the upper villi at 20 w of SAMP1/YitFc mice have been added to Fig 1A.

1-2: Figure 1f: The authors should split the channels to better show the expression of MUC2 and Crp1.

Separated channel images for Muc2 and Crps are now added to Fig 1F.

1-3: Figure 2: The western-blot images are not very convincing. It is difficult to determine which bands have been quantified (Especially for pIRE1 and GRP78). The quantifications (fig 2d) do not show standard error bars.

To perform appropriate statistical processing, including the description of error bars, the number of experiments in each group was increased from two to four, and new western blot analyses were performed (Fig 2F), and the results are now included in the revised manuscript text at line 149–154 and new figure, Fig 2G. In addition, the blocking conditions were changed from 1 hour at room temperature to overnight at 4 °C to reduce membrane background and clarify bands for quantification (line 477). As a result, although pIRE1 α levels showed no statistical difference between 4 w and 20 w, levels of ATF4, cleaved ATF6, and GRP78 ER stress markers in ileal crypts increased significantly in SAMP1/YitFc mice at 20 w compared to those at 4 w along with the disease progression (Fig 2G). We appreciate the opportunity to strengthen our manuscript in response to your valuable comments in this revision.

1-4: Figure 3: What do “fraction” stands for? Does that mean different part of the small intestine? Please precise. Please also precise the number of experiment performed in the legend.

We apologize for the confusion created by our original manuscript. In Fig 3, we used continuous elution electrophoresis to separate proteins extracted from the entire length of each small intestine collected from ten ICR and ten SAMPI/YitFc mice. The fraction obtained was used for western blot analysis. Fraction numbers correspond to the order of elution during the electrophoretic separation. To describe this point more clearly, we have added the number of mice (10 each) used to the legend of Fig 3 (line 918-919), and we have revised the methods (line 507-515) to better explain the experimental approach.

1-5: Supplementary figure 1: Please label the westerns/dot blots. It is not possible to understand what is shown on the figures as currently presented.

Appropriate labels have been added to the dot blot images (previously, supplementary Fig 1a; now, Fig S6A in the revision), and the figure legend of Fig S6A was modified to clarify the purpose and conclusions of the experiment. Related to the concern, we also modified CBB images adding dotted line for distinguishing between oxCrps and rCrps (Fig S6B).

1-6: line 60: “suggests show”

We corrected line 63 (moved from line 60) from “suggests shows” to “shows”.

1-7: line 215: On the graphs, it appears that Bacteroides was positively correlated to inflammatory score and Anaerotruncus was negatively correlated to inflammatory score. The text should be corrected accordingly.

The text of line 249–251 (moved from line 214–215) was corrected to read “Moreover, at the genus level, relative abundance of *Lachnospiraceae;Other* and *Anaerotruncus* correlated negatively and *Bacteroides* correlated positively with the inflammatory score”.

1-8: line 230-232: Please soften the conclusion: “could be involved” instead of “is involved”.

As suggested, we softened the tone of the conclusion, from “is involved” of line 301 (moved from line 232) to “could be involved”.

1-9: Line 234-236: The sentence is not clear.

The text of line 305–308 (moved from line 234–236) was modified to achieve better clarity to read “Dysbiosis with reduced diversity observed in the SAMP1/YitFc mice is consistent with previous studies of the intestinal microbiota of CD patients from the America, Europe, and Japan. Furthermore, decreases of

both Lachnospiraceae and Ruminococaceae along with disease progression shown here also have been reported in CD patients in the America and Japan.”.

1-10: The English language should be checked by a native English speaker as many errors remain in the text.

The entire manuscript was carefully examined and checked by the native speaker. Thank you for your constructive advice.

Response to Referee #2

Thank you for constructive suggestions and comments for further improvement of our revised manuscript. As recommended, we performed additional experiments and have updated the revised manuscript by incorporating new findings. The following is our point-by-point response.

2-1: Authors claim significant crypt elongation during progression of ileitis in their mouse model - please quantify this. The specified crypt abscess is also not clear from Figure 1.

All data including crypt elongation, inflammatory cell counts, muscle layer thickness, and villus length used to calculate inflammatory scores shown in Fig 1A have now been added as new Fig. S1A–D. In addition, a representative image of a crypt abscess in a 20 w SAMP1/YitFc mouse has been added as new Fig. S1E.

2-2: Reduction in the size of Paneth cell granules is clear - does the number of granules per cell also change during disease progression? Do the dysfunctional Paneth cells still express lysozyme?

To address these concerns, we added new Fig 2C–E comparing the number of granules per Paneth cell in SAMP1/YitFc mice between 4 w and 20 w from TEM images (Fig S4). The data show that the

granule area of Paneth cells did not change between 4 w and 20 w (Fig 2C) in SAMP1/YitFc mice, but the number of granules increased significantly at 20 w compared to 4 w (Fig 2D). To further quantify abnormalities in the ER of Paneth cells associated with the pathological progression of SAMP1/YitFc mice, ER lumen diameter was quantified, and it was increased significantly at 20 w compared to 4 w in SAMP1/YitFc mouse ileal tissues (n = 3 each). The method for Fig 2C-E was added to the method section (line 459–464), and the results are now included in the revised manuscript text at line 143–147.

In Fig A for referee #2, we showed the results of Crps and lysozyme co-localized immunostaining in the ileum of 20 w ICR and SAMP1/YitFc mice. As is evident, 20 w ICR mice expressed lysozyme in all Paneth cells at the base of the crypt. In SAMP1/YitFc mice, although Crp-positive Paneth cells at the base of the crypt stained for lysozyme, abnormal Paneth cells in the upper crypts and on villi do not express lysozyme. These findings suggest that unlike cryptdins, the expression and localization of lysozyme in SAMP1/YitFc mice may be regulated normally.

Figure A for referee #2. Expression of lysozyme in abnormal Paneth cells.

Representative immunofluorescent staining images for lysozyme (green) in ileal tissues. Crps (red) and DAPI (blue). Scale bars indicate 20 μ m.

2-3: Presence of altered Paneth cells in upper crypts and villi not clear from figure 1. What causes this? Dysregulated Wnt signaling causing loss of EpH/ephrin boundaries?

To address these concerns, we added arrows that indicate abnormal Paneth cells in the upper crypts and villi to Fig 1A. To address possible mechanisms of abnormal Paneth cells in the upper crypts and villi, the expression of EphB2, which defines the position of Paneth cells at the base of the crypt (PMID: 12408869), was analyzed by immunofluorescent staining and added as Fig S2. The results were included in the revised manuscript text at line 125–128. Along with disease progression, EphB2 expression was reduced in abnormal Paneth cells in the crypts. No EphB2 expression was detected in abnormal Paneth cells on villi of 20 w SAMP1/YitFc mice. Taken together, these new results suggest that the decreased expression of EphB2 that accompanies disease progression involve the transfer of abnormal Paneth cells onto villi.

2-4: Are the defective Paneth cells found throughout the intestines? If so, why does disease typically present in the terminal ileum? Is the Paneth cell phenotype seen in other mouse models of Crohn's disease (eg, TNFalpha-overexpressing mice)?

In our study, development of enteritis and abnormalities of Paneth cells are localized in the ileum, especially in the terminal ileum. Mechanisms of disease progression and why abnormalities of Paneth

cells are localized in the terminal ileum in SAMP1/YitFc mice remain unknown. GWAS analyses revealed Crohn's disease patients that develop ileal disease have a higher percentage of dysfunctional NOD2 polymorphisms compared to those with no ileal disease (PMID: 26490195). In SAMP1/YitFc mice, although NOD2 itself has no genetic variation, its response to MDP ligands is reportedly much lower than in wild-type mice (PMID: 24082103). Thus, dysregulation of innate immune responses might contribute to ileum-specific pathogenesis in SAMP1/YitFc mice.

In the TNF α Δ ARE mice that the referee pointed out, no ectopic appearance of Paneth cells has been reported. Also, the reduction of Paneth cells in the ileum (PMID: 26487367) was reported in the TNF model, suggesting that pathogenesis differs from SAMP1/YitFc mice. Other mouse models showing Paneth cell abnormalities include the knockout mice of AGR2, an ER chaperone and a Crohn's disease susceptibility gene (PMID: 20025862), as well as IRGM1 knockout mice, a GTPase involved in autophagy induction (PMID: 23989005). AGR2 knockout mice reportedly develop enteritis from the terminal ileum to the large intestine, and ileal Paneth cells are found not only at the base of the crypts but also in the upper crypts and on villi. In addition, the expression of ER stress markers in the small intestine is increased. IRGM1 knockout mice have abnormal lysozyme/Muc2-double positive, small granules in ileal Paneth cells. In addition, IRGM1 knockout mice have been known to show increased numbers of Paneth cells and when dextran sulfate sodium (DSS) was administered, inducing not only colitis but also ileitis which is not normally seen with DSS administration. Since both of these genes function to remove abnormal proteins in cells, we think that

impaired protein homeostasis due to enhanced ER stress that we have observed is related to Paneth cell abnormalities in SAMP1/YitFc mice. However, the detailed mechanisms remain to be determined.

2-5: Co-expression of Muc2 and Crps appears restricted to the upper crypts/villi rather than the Paneth cell zone at the crypt base - this suggest that goblet cells might be converting into Paneth cells rather than vice versa - could these altered goblet cells be contributing to disease progression?

To address this important question, we first added separate channel images for each molecule to clarify the localization of Crps and Muc2 in Fig 1F. As a result, abnormal Paneth cells co-expressing Muc2 and Crps were observed from the crypt base to upper crypts/villi. In addition, As described in the point-by-point response letter to the referees [2-3], the expression of EphB2 defining the position of Paneth cells was reduced along with disease progression (Fig S2), suggesting that the abnormal Paneth cells are due to Paneth cell failure. These results support the view that abnormal localization of Paneth cells in the upper crypts and on villi suggest that Paneth cells may acquire goblet cell traits while migrating from the crypt base to the villus.

Perhaps, goblet cells were converted to Paneth cells since Muc2 expression was higher in abnormal Paneth cells in the upper crypt and the villi, and the presence of this “altered goblet cell” may be involved in the pathophysiology of SAMP1/YitFc mice. However, details are unknown. We would like to address the issue in our future research. Thank you for your important suggestions.

2-6 (1): Figure 2c - Western blots are not publication quality - It is very difficult to concur with the claims of altered ATF4/IRE expression based on these data. Why not use ER stress markers in IHC/IF? What about IRE1B and downstream targets of Xbp1 (via q-PCR)? It is also questionable whether the use of whole crypt extracts allows conclusions regarding ER stress specifically in Paneth Cells - multiple cell types in the crypts may be undergoing ER stress.

To address your concerns, additional experiments were performed with the improvement of the method. As described in the point-by-point response letter to the referees [I-3], we obtained the clear data that significant increase of ER stress marker expression is observed in 20 w SAMP1/YitFc mice, and the results were added in Fig 2F and G.

To determine whether ER stress in the crypts that was detected by western blot occurs in Paneth cells, ileal tissue sections from both ICR and SAMP1/YitFc mice were tested by immunofluorescent staining for expression of the ER stress markers GRP78 and calreticulin, which are downstream of ATF6, a representative ER stress sensor and MIST1 (PMID: 31330316), a transcription factor which is involved in suppression of ER stress and is expressed specifically in Paneth cells were newly performed and added in figure Fig S5. The results were included in the revised manuscript text at line 158–164. In Paneth cells of normal 20 w ICR mice, GRP78 and calreticulin were both present at low levels. In contrast, the expression of GRP78 and calreticulin were both increased in abnormal Paneth cells of SAMP1/YitFc mice but not in other epithelial cell lineages at 20 w, indicating that ER stress

response occurred mainly in SAMP1/YitFc mouse Paneth cells. In addition, MIST1 was abundant at high levels in Paneth cell nuclei of ICR mice. In sharp contrast, MIST1 expression in SAMP1/YitFc mouse Paneth cells was dramatically reduced at 20 w compared to 4 w (Fig S5C). Taken together, we confirmed that ER stress in crypts of SAMP1/YitFc mice occurs in Paneth cells. We appreciate this valuable suggestion, which has allowed our revised manuscript to be improved and strengthened.

2-6 (2): Why not purify the Paneth cells from the normal and diseases intestines (for example using the CD24hi/SSChi protocol from Sato et al) for more specific analyses of the changes to Paneth cells? This would also facilitate a more unbiased approach to evaluating changes in the Paneth cells during disease progression.

Sorting Paneth Cells from intestinal organoids derived from human Crohn's disease patients would also facilitate validation of the major findings and establish the relevance for human disease.

We agree that isolating Paneth cells would enable Paneth cell-specific analyses. Thus, first we stained the ileum of SAMP1/YitFc mice with anti-CD24 and confirmed that the CD24 expression is low in abnormal Paneth cells (data not shown), which eliminated the prospect of cell sorting approaches. Therefore, to clarify the issue raised the referee, alternatively, we conducted fluorescent immunostaining of SAMP1/YitFc mouse ileum for ER stress markers and have shown that ER stress occurs in abnormal Paneth cells. These new data were added to the revised manuscript text in results

and new Fig S5 as we responded to item **2-6 (1)**. In response to this constructive suggestion, we have confirmed that ER stress occurs in abnormal Paneth cells.

2-6 (3): Sorting Paneth Cells from intestinal organoids derived from human Crohn's disease patients would also facilitate validation of the major findings and establish the relevance for human disease.

We hope that you will agree that this fascinating suggestion is outside the scope of this study. This study experimentally demonstrated that ER stress in abnormal Paneth cells causes secretion of reduced-form α -defensin using a Crohn's disease model mouse SAMPI/YitFc. We are very much interested in your view, and it will be one of important targets in our future study. Thank you very much for your valuable suggestion.

2-7: Are misfolded forms of Defensins present in human fecal samples from Crohn's disease patients?

The application of our findings to actual diagnosis or treatment of Crohn's disease patients is an eventual goal. However, in the absence of confirming preliminary data, we believe that prospect is

outside the scope of this paper but an important consideration for future studies. Thank you for your understanding.

2-8: Discussion needs to be much more concise and presenting detailed discussion rather than an extended re-iteration of the findings.

Thank you for the comment. We revised Discussion to be more concise as reducing three lines with adding much detailed discussion as you indicated.

Response to Referee #3

Thank you for constructive suggestions and comments for further improvement of our revised manuscript. Based on those comments and concerns, we have conducted substantial additional experiments and updated the revised manuscript. The following is our point-by-point response to your comments.

3-1: Crohn's disease (CD) patients are known to have reduced defensin expression in Paneth cells (PMID: 19904243), and yet in this study, SAMP1/YitFc mice have increased Paneth cells and cryptdin expression. The discrepancy calls into the question the relevance as a "CD" mouse model.

It is well known that the etiology of Crohn's disease is heterogeneous and that its pathology varies greatly between patients. Indeed, Wehkamp *et al.* have reported that decreased HD5 expression occurs in Crohn's disease patients who carry NOD2 mutations (PMID: 15479689). On the other hand, it also has been reported that the decrease in HD5 expression level in Crohn's disease patients is secondary to Paneth cell deletion by severe intestinal inflammation (PMID: 18305068). Also, protein expression of HD5 showed significantly higher levels in moderate and severe Crohn's disease patients compared to ulcerative colitis patients (PMID: 28817680). Thus, the relation between Crohn's disease and

Paneth cell α -defensin expression levels remain inconsistent if not controversial, and we think that future research is needed to resolve this question.

The SAMP1/YitFc mice we used in this study is a CD model mouse that spontaneously develops ileitis with pathology that includes inflammatory cell infiltration, muscle layer thickening, and granuloma formation without any chemical or genetic induction, which is similar to CD patients. In addition, it has been reported that point mutations accumulate in chromosomal 9 loci containing CD susceptibility genes such as IL10R α and IL18 in SAMP1/YitFc mice (PMID: 21557393). Additional similarities of the SAMP1/YitFc mice to CD patients include involvement of dysbiosis in its onset (PMID: 17237431), incidence of extraintestinal manifestations, and the improvement of the disease state by administration of TNF α antibody used for clinical treatment and other conventional therapies (PMID: 12832622). Based on these previous reports, we think that SAMP1/YitFc mouse is a suitable preclinical model for analyzing the pathogenesis and pathophysiology of CD.

3-2: While Paneth cell defects can lead to dysbiosis, an alternative hypothesis is that the microbes trigger Paneth cell abnormality. Examples of how environmental factors can affect Paneth cells include PMID: 20602997, 30137026. To exclude that dysbiosis could lead to Paneth cell defects, an experiment can be set up where mice are given antibiotics to deplete these microbes, and determine if Paneth cells remain abnormal. Alternatively, if the authors have these mice bred in gnotobiotic facility, that will also provide tremendous insight.

To address the alternative hypothesis that has been raised, we conducted additional experiments to test whether the intestinal microbiota may induce Paneth cell abnormalities in SAMP1/YitFc mice. We analyzed the morphology of Paneth cells and secretion of rCrps after oral administration of antibiotics to SAMP1/YitFc mice. The results were added as Fig S12 and included in the revised manuscript text at line 274–283. Following a previously reported protocol (PMID: 21445311), SAMP1/YitFc mice were administered a cocktail of antibiotics (Abx) orally from 4 to 10 weeks of age (n = 3, Fig S12A). PCR analyses of fecal 16S rDNA confirmed that intestinal bacteria were eliminated completely by antibiotic treatment for 6 weeks (Fig S12B). At 10 w, the number of eosinophilic granule positive cells, i.e., abnormal Paneth cells, per villus-crypt axis in Abx-treated group was unchanged with no statistical difference from untreated mice. Because the number of mice that could be studied during this period under revision was limited, the data in Fig 1D were used again as controls for statistical processing. (Please see Fig B for referee #3 and Fig S12C.). The number of eosinophilic granule positive cells in a water-treated mouse (n = 1), to which distilled water was administered, was at the same level as in controls. TEM analyses revealed that abnormal granule morphology and ER swelling are shown in Paneth cells of both the Abx group and water-treated mouse (Fig S12D). Furthermore, rCrps were secreted into the feces of both the Abx group and the water-treated mouse (Fig S12E). These results indicated that Paneth cells are abnormal even in the absence of the intestinal bacteria during the pathogenesis of SAMP1/YitFc mice.

Figure B for referee #3. Paneth cell abnormalities after antibiotics treatment in SAMP1/YitFc mice.

Number of eosinophilic granule number. Data plotted in Fig 1D were reused as Control, and error bars represent mean \pm SEM. Statistical significance was evaluated by Mann-Whitney's U test. $p < 0.05$ was considered statistically significant.

3-3: Along the same line, to claim that the excessive reduced-form cryptdin alters microbiota composition, the authors need to show that supplement with reduced-form cryptdins can alter the microbiota in wild type mice.

In response, we performed additional experiments to test whether rCrps cause dysbiosis by administrating the rCrp1 to normal ICR mice and the intestinal microbiota composition was analyzed. The results were added as new figure, Fig S13, and included in the revised manuscript text at line 288–298. We administered rCrp1 rectally, since oral administration of rCrps may result in degradation in the stomach and loss of activity in the intestinal tract, and rCrps could be degraded by proteolysis (PMID: 15297466). After habituation for 1 w, rectal administration (day 1, 3) and feces collection (day 0, 2, 4) for the intestinal microbiota analysis were performed every other day (Fig S13A). No difference was observed in α -diversity of the intestinal microbiota between the rCrp1 group and the

control group at day 0. At day 4, the observed OTUs which indicate α -diversity in the rCrp1 group were significantly decreased compared to those in the control group (Fig S13B). Furthermore, Lachnospiraceae were significantly decreased in the rCrp1 group at day 4 and Ruminococaceae (shown in Fig 5C) were significantly decreased in the rCrp1 group at day 2 and day 4, both which were decreased along with disease progression of SAMP1/YitFc mice (Fig S13C). In addition, abundances of Ruminococaceae showed significant negative correlation with the inflammatory score and the amount of rCrps secretion (Fig S14). These new results indicated that administration of rCrp1 to normal ICR mice can induce dysbiosis partially reproducing the dysbiosis found in SAMP1/YitFc mice. Taken together, we have now confirmed that supplementation with rCrps can alter the microbiota in wild type mice. These new findings strengthen our paper in response to the suggestions of the referee.

3-4: The authors' claim that altered cryptdins are being "secreted" into the intestinal lumen lacks experimental proof. The Paneth cells may simply die of excessive ER stress and as a result, the cryptdins are being released into the lumen. There is no data to define Paneth cell survival, turnover, cell death, nor a proper in vivo secretion assay (one such example would be PMID: 18849966).

To address the referee's concerns, we conducted additional experiments to confirm that Crps are secreted from Paneth cells in SAMP1/YitFc mice. Using enteroids, a three-dimensional culture system of small intestinal epithelial cells including Paneth cells, Paneth cell survival, cell death, and granule secretion were analyzed. The results were added as new figure, Fig S7 and new Video1 and included in the revised manuscript text at line 220–230. Paneth cells in enteroids prepared from isolated crypts from SAMP1/YitFc mice had significantly smaller granules than those of ICR mice as observed *in vivo* (Fig S7A and B). These results indicated that enteroids are suitable for evaluating abnormal Paneth cells of SAMP1/YitFc mice. Next, we tested whether SAMP1/YitFc mouse-derived abnormal Paneth cells have the ability to secrete granules. When carbachol (CCh), which induces Paneth cell granule secretion, was added to the culture medium, Paneth cells of SAMP1/YitFc mouse enteroids secreted granules equivalent to secretion by Paneth cells in ICR mouse enteroids (Fig S7B). We further tested whether abnormal Paneth cells survived during and after induced secretion, i.e., did not undergo induced cell death. After treatment of SAMP1/YitFc mouse enteroids with a fluorescent probe specific for active caspase-3/7, a method that we reported previously (PMID: 30232288), granule secretion was induced by CCh. No cleaved caspase was detected during or after secretion by Paneth cells (Fig S7C and Video 1). Taken together, we conclude that SAMP1/YitFc mouse Paneth cells possess secretory abilities and do not undergo apoptosis even after secretion. These results confirm that the abnormal Paneth cells of SAMP1/YitFc mice secrete rCrps.

3-5: It is known that the genetics of the SAMP1/YitFc mice is unclear. As a result, the use of ICR mice (with no description of background...etc) as a control for microbiome studies is suboptimal.

What is the baseline difference of microbiota compositions between SAMP1/YitFc and ICR mice?

This data should be included. The authors should also include PCoA plot analyses for 0, 4, and 20 weeks time points.

To address the referee's concerns, we conducted additional analysis. To compare the intestinal microbiota of ICR and SAMP1/YitFc mice, β -diversity was analyzed using PCoA plot at 4 w, the starting point of the study and the youngest mice available for use, and at 20 w, the end point of the study. The results were added as new figure Fig S8 and included in the revised manuscript text at line 251–252. There was no significant difference in β -diversity between ICR and SAMP1/YitFc mice at 4 w. In contrast, at 20 w when disease has progressed in SAMP1/YitFc mice, the composition of the intestinal microbiota in these groups were significantly different (Fig S8). Thus, SAMP1/YitFc mouse intestinal microbiota is similar to that of ICR mice at baseline before the onset of disease.

3-6: Many key data lack quantification: For example, how often did the authors observe ER changes in TEM in Figure 2? From previous study (PMID: 15793286), this may be a very small proportion.

To address the referee's comments about how often Paneth cells with ER abnormalities were found, we conducted additional experiments, and these data are now included in the revised manuscript text at line 141–142. Two crypts were randomly selected from 20 w SAMP1/YitFc mice, and the ER was observed in three Paneth cells, i.e., a total 18 Paneth cells. A representative field of view is shown as new supplementary figure, Fig S3. As a result, ER swelling and fragmentation were observed in all 18 Paneth cells, indicating that abnormalities in the ER occurred in almost all 20 w SAMP1/YitFc mouse Paneth cells.

As described in the point-by-point response letter to the referees [2-2], we addressed your concerns by performing additional experiments and analyses. To quantify abnormalities in the ER of Paneth cells associated with the pathological progression of SAMP1/YitFc mice (new figures, Fig 2C–E), TEM images were obtained from 4 w and 20 w SAMP1/YitFc mouse ileal tissues (new supplementary figure, Fig S4), and the method was added to the method section at line 459–464 and the results are now included in the revised manuscript text at line 143–147. As can be seen, the granule area of Paneth cells did not change between 4 w and 20 w (Fig 2C), but the number of granules increased significantly at 20 w compared to 4 w in SAMP1/YitFc mice (Fig 2D). In addition, the ER inner diameter (ER lumen diameter) was significantly increased at 20 w compared to 4 w (Fig 2E). The site measured the ER lumen diameter of the Paneth cells was shown as Fig C for referee #3 below, with red lines highlighted.

Figure C for referee #3. ER stress in abnormal Paneth cells.

Representative TEM images of Paneth cells at the base of ileal crypts in SAMPl/YitFc mice. Red lines represent ER lumen diameter. Scale bars indicate 1 μm .

3-7: 1. The authors should provide rationale for testing cryptdin 4 but rather than the more Paneth cell-specific cryptdin 4.

In this study, we analyzed using pan-cryptdin antibody that reacts with cryptdin-1, 2, 3, 4, and 6, which are specifically expressed in Paneth cells, including cryptdin-4 (Fig S6A). Based on the referee's comment, a more clear explanation has been added to the legend of Fig S6 (line 985–989).

3-8: Figure 1: this is essentially confirmatory of what has been previously published (PMID: 15793286), yet in figure 1f there seems to be very few overlap staining between cryp1 and muc2.

To address the referee's concern, separate channel images of Crps and Muc2 have been now added in Fig 1F and show the expression of each molecule more clearly. As can be seen, all Crp-positive cells in SAMP1/YitFc mice at 20 w were Muc2-positive, consistent with the PAS-alcian blue staining shown in PMID: 15793286. Thank you for the point of clarification.

3-9: Likewise, it was not clear from Fig.1a how the Paneth cell granule sizes are different between the two strains of mice. The authors need to provide methodology how they measured the granule size (how many granules, how many Paneth cells were counted), as well as a blown-up picture to highlight the size difference.

According to the referee's comments, the method for calculating the particle size in Fig 1A was added to the Method section (line 433–437). As you can see, all granule diameters were measured for a total of 9 Paneth cells (3 Paneth cells per crypt). Fig D for referee #3 shows representative ileal Paneth cells at high magnification of HE from ICR and SAMP1/YitFc mice that were used for the actual measurements. We also conducted detailed analysis for granules of abnormal Paneth cells using TEM images as described in the point-by-point response letter to the referees [3-6].

Figure D for referee #3. HE staining images used for granule diameter measurement.

Representative HE staining images of Paneth cells at the base of ileal crypts in 20 w ICR and SAMP1/YitFc mice. Black straight lines represent the maximum diameter of each Paneth cell granule measured.

3-10: In Fig. 2. the authors only showed 2 mice each for each of the panel. This is insufficient for statistics.

According to your suggestions, additional western blots increasing the number of experiments in each group from two to four were performed, and the results sufficient for statistical analyses are now included in the revised manuscript text at line 149–154. Thank you for your valuable comments. Your comments have strengthened our revised manuscript.

3-11: The use of the term “reduced cryptdins” may be confusing for some (as compared to reduced quantity of crypidins). Suggest to change it (such as reduced-form cryptdins) to avoid such confusion.

As you kindly suggested, the terms “reduced” and “oxidized” have been changed to “reduced-form” and “oxidized-form”.

3-12: Figure 5, legend: there does not seem to have ICR mice data in this figure.

As noted, description of ICR mice was removed from the legend in Fig 5A and 5B (line 942).

3-13: Supplementary figure 1 is incomprehensive as there was no label to describe these blots.

We apologize for lack of appropriate information in the dot blot images. As described in the point-by-point response letter to the referees [I-5], we corrected the point you suggested in Fig S6A.

March 26, 2020

RE: Life Science Alliance Manuscript #LSA-2019-00592-TRR

Prof. Tokiyoshi Ayabe
Hokkaido Univ
Faculty of Advanced Life Science
Kita-21, Nishi-11, Kita-ku
Sapporo 001-0021
Japan

Dear Dr. Ayabe,

Thank you for submitting your revised manuscript entitled "Paneth cell α -defensin misfolding correlates with dysbiosis and ileitis in Crohn's disease model". The reviewers re-evaluated your work and their comments are attached below. We would be happy to publish your paper in Life Science Alliance pending the following final revisions:

- please respond to the remaining comments of rev#2 and #3; I would recommend including the CD24 staining (rev#2) in your point-by-point response, to respond to point 1 of reviewer #3 and to extend the discussion accordingly, and to remove data & statements on Paneth cell granule size
- the callout to figure S7C is currently in the legend of a supplementary movie, please mention it also when referring in the manuscript text to that movie.
- please deposit the 16S rDNA metagenomic sequencing results in a repository and provide in the M&M data availability section the accession code to those.

A. FINAL FILES:

B. MANUSCRIPT ORGANIZATION AND FORMATTING:

Thank you for your attention to these final processing requirements.

Sincerely,

Reviewer #1 (Comments to the Authors (Required)):

Dear Editor,

The paper by Shimizu et al has now been modified according to my suggestions.

It is important to note that the paper is now of great quality, due to many experimental data added to the result section and to the enriched discussion.

This is a very interesting paper which represents a real advance in the field of IBD research with strong data, proper quantifications and interesting experimental approach.

Kind regards

Reviewer #2 (Comments to the Authors (Required)):

Many technical concerns have been alleviated and the discussion improved.

Given that sorting of Paneth cells from their disease model would have facilitated an unbiased analysis of temporal changes occurring, I am surprised that the authors did not show clear data supporting a reduction in CD24 levels in the Paneth cells over time to substantiate their claim that sorting using the Sato method was not feasible.

Reviewer #3 (Comments to the Authors (Required)):

The authors have conducted extensive experiments for the revision and should be lauded. These work addressed most of the comments made previously, with the two areas unresolved. I will recommend that, should the paper be considered for acceptance, the authors remove the related statements or significantly modify their claims and include in discussion that the areas that are unsolved.

1. The relationship between microbiota composition, genetics, and administration of reduced form Crp1: My interpretation of these data is contrary to the statement made by the authors. The authors showed that at 4 weeks, there is no significant changes between ICR and SAMP1 mice (Fig. S8). The difference became significant only at 20 weeks. However, in Fig. S13, 4 days treatment of reduced form Crp 1 reduces alpha diversity and this has been claimed by the authors of evidence that it reproduced the microbiota changes seen in the SAMP1 mice. These data not only are not comparable but the conclusion misleading. How is it possible it takes 16 weeks for mouse models to show dysbiosis and yet with administration, such effect can be seen in 4 days? PCoA plots need to be shown. The Lachnospiraceae data in panel C seems to be due to 1 single outlier. I would suggest removing the microbiota data altogether as this is sending mixed message.
2. Paneth cell size: In Figure 1C and F, the granule size in ICR mice at 20 weeks in Fig 1F does not support the quantification shown in Fig 1C. Also, from Fig S3, it appears there is a wide spectrum of granule size in these mice, which does not seem to be supported by the data in Fig 1C. Also, the authors quantified 3 Paneth cells per crypt, and 3 crypts per mouse. While n=3 appears to be the minimal number by certain statistical standards, the other groups that have already made contributions in this area have counted 40->2000 crypts per mouse. Given the sampling issue and unresolved variation issue, I would recommend removing all data and claims about Paneth cell size.

To the Editor:

Thank you for your directions. We agree with and followed all of your constructive suggestions.

- please respond to the remaining comments of rev#2 and #3; I would recommend including the CD24 staining (rev#2) in your point-by-point response, to respond to point 1 of reviewer #3 and to extend the discussion accordingly, and to remove data & statements on Paneth cell granule size

According to your direction, we responded the remaining comments of rev#2 and #3. In addition, to respond to the point 1 of rev#3, we included additional statements explaining why the SAMP1/YitFc mouse is suitable model for Crohn's disease in Discussion section, line 311–315, “In addition, SAMP1/YitFc mice show the improvement of the disease state by administration of TNF α antibody used for clinical treatment and other conventional therapies [58]. Taken together, it is suggested that SAMP1/YitFc mouse is a suitable preclinical model for analyzing the pathogenesis and pathophysiology of CD.” and added a new reference [58] in Reference section.

- the callout to figure S7C is currently in the legend of a supplementary movie, please mention it also when referring in the manuscript text to that movie.

We corrected this as mentioning Fig S7B (moved from Fig S7C).

- please deposit the 16S rDNA metagenomic sequencing results in a repository and provide in the M&M data availability section the accession code to those.

We deposited the 16S rDNA metagenomic sequences in the Materials and Methods section as Data availability.

To the reviewers:

Thank you for the reviewers' constructive comments and suggestions on our revised manuscript.

Response to Reviewer#1:

The paper by Shimizu et al has now been modified according to my suggestions.

It is important to note that the paper is now of great quality, due to many experimental data added to the result section and to the enriched discussion.

This is a very interesting paper which represents a real advance in the field of IBD research with strong data, proper quantifications and interesting experimental approach.

Thank you for your valuable comments and suggestions which enabled the revised manuscript to be improved.

Response to Reviewer#2:

Many technical concerns have been alleviated and the discussion improved.

Given that sorting of Paneth cells from their disease model would have facilitated an unbiased analysis of temporal changes occurring, I am surprised that the authors did not show clear data supporting a reduction in CD24 levels in the Paneth cells over time to substantiate their claim that sorting using the Sato method was not feasible.

Our preliminary data showed negative expression of CD24 in SAMP1/YitFc mouse Paneth cells in ileum as you can see at Figure for Reviewer#2. Paneth cell sorting would have proven too difficult, substantiated by the fact that preliminary anti-CD24 staining showed no significant staining of Paneth cells. More importantly, though, the CD24 staining issue is tangential to the primary scope of the paper, since we have demonstrated that ER stress occurs specifically in abnormal Paneth cells of SAMP1/YitFc mice by both IHC and TEM analyses. For these reasons, we would prefer to exclude CD24 statements regarding CD24 from the manuscript. We appreciate your understanding.

Figure for Reviewer#2. Negative expression of CD24 in abnormal Paneth cells of SAMP1/YitFc mice.

Immunofluorescent staining images for CD24 (green) using anti-mouse CD24 antibody (M1/69, Biolegend) and Crps (red) in ileal tissues of ICR mouse and SAMP1/YitFc mouse were shown. Nuclei (blue). Scale bars: 20 μ m.

Response to Reviewer#3:

The authors have conducted extensive experiments for the revision and should be lauded. These work addressed most of the comments made previously, with the two areas unresolved. I will recommend that, should the paper be considered for acceptance, the authors remove the related statements or significantly modify their claims and include in discussion that the areas that are unsolved.

1. The relationship between microbiota composition, genetics, and administration of reduced form Crp1: My interpretation of these data is contrary to the statement made by the authors. The authors showed that at 4 weeks, there is no significant changes between ICR and SAMPI mice (Fig. S8). The difference became significant only at 20 weeks. However, in Fig. S13, 4 days treatment of reduced form Crp1 reduces alpha diversity and this has been claimed by the authors of evidence that it reproduced the microbiota changes seen in the SAMPI mice. These data not only are not comparable but the conclusion misleading. How is it possible it takes 16 weeks for mouse models to show dysbiosis and yet with administration, such effect can be seen in 4 days? PCoA plots need to be shown. The Lachnospiraceae data in panel C seems to be due to 1 single outlier. I would suggest removing the microbiota data altogether as this is sending mixed message.

We apologize for the mixed message and confusion that our unclear statement caused. Our objective was to test whether rCrp could modulate the fecal microbiota. As a result, the fecal enema of rCrp to ICR mice did not replicate the *in vivo* SAMPI/YitFc mouse enteropathy. However, we found that rCrp modulate the intestinal microbiota. According to the reviewer's constructive comments, we changed manuscript text to clarify the objective and the results of the experiments as follows.

Line 275–276: “Finally, we further tested whether reduced-form Crps could change the intestinal microbiota using ICR mice.”

Line 288–290: “Although rectal administration of rCrp to ICR mice did not replicate the SAMPI/YitFc mouse enteropathy, we found that rCrp could modulate the fecal microbiota.”

2. Paneth cell size: In Figure 1C and F, the granule size in ICR mice at 20 weeks in Fig 1F does not support the quantification shown in Fig 1C. Also, from Fig S3, it appears there is a wide spectrum of granule size in these mice, which does not seem to be supported by the data in Fig 1C. Also, the authors quantified 3 Paneth cells per crypt, and 3 crypts per mouse. While n=3 appears to be the minimal number by certain statistical standards, the other groups that have already made contributions in this area have counted 40->2000 crypts per mouse. Given the sampling issue and unresolved variation issue, I would recommend removing all data and claims about

Paneth cell size.

We fully understand and follow the referee's suggestion. We now remove all data regarding Paneth cell granule size as the referee recommended. The corrections include line 91–92, line 107 (previous line 108-112 removed), line 131-132 (moved from line 135-137), line 139-140 (moved from line 143-146), line 214 (previous line 220-223 removed), line 270-271 (moved from line 279-281), line 327-330 (moved from line 334-337), line 426 (previous line 433-437 removed), line 519 (previous line 530-531 removed), line 894 (previous line 898-899 removed), line 904-906 (moved from line 909-911), and line 992 (previous line 998 removed).

April 7, 2020

RE: Life Science Alliance Manuscript #LSA-2019-00592-TRRR

Prof. Tokiyoshi Ayabe
Hokkaido Univ
Faculty of Advanced Life Science
Kita-21, Nishi-11, Kita-ku
Sapporo 001-0021
Japan

Dear Dr. Ayabe,

Thank you for submitting your Research Article entitled "Paneth cell α -defensin misfolding correlates with dysbiosis and ileitis in Crohn's disease model". I appreciate the introduced changes and it is a pleasure to let you know that your manuscript is now accepted for publication in Life Science Alliance. Congratulations on this interesting work.

DISTRIBUTION OF MATERIALS:

Again, congratulations on a very nice paper. I hope you found the review process to be constructive and are pleased with how the manuscript was handled editorially. We look forward to future exciting submissions from your lab.

Sincerely,

Andrea Leibfried, PhD
Executive Editor
Life Science Alliance
Meyerohofstr. 1
69117 Heidelberg, Germany
t +49 6221 8891 502
e a.leibfried@life-science-alliance.org
www.life-science-alliance.org